# Teaching Invariance Using Privileged Mediation Information

**Dylan Zapzalka**  *dylanz@umich.edu*
*Computer Science and Engineering Division*
*University of Michigan*

**Maggie Makar**  *mmakar@umich.edu*
*Computer Science and Engineering Division*
*University of Michigan*

**Reviewed on OpenReview:** *https: // openreview. net/ forum? id= 8ZLhuo32Kz*

## Abstract

The performance of deep neural networks often deteriorates in out-of-distribution settings due to relying on easy-to-learn but unreliable spurious associations known as shortcuts. Recent work attempting to mitigate shortcut learning relies on *a priori* knowledge of the shortcuts and invariance penalties, which are difficult to enforce in practice. To address these limitations, we study two causally-motivated methods that efficiently learn models that are invariant to shortcuts by leveraging *privileged mediation information*. We first adapt concept bottleneck models (CBMs) to incorporate mediators — intermediate variables that lie on the causal path between input features and target labels — resulting in a straightforward extension we call Mediator Bottleneck Models (MBMs). One drawback of this method is that it requires two potentially large models at inference time. To address this issue, we propose Teaching Invariance using Privileged Mediation Information (TIPMI), a novel approach which distills knowledge from a counterfactually invariant teacher trained using privileged mediation information to a student predictor that uses non-privileged, easy-to-collect features. We analyze the theoretical properties of both estimators, showing that they promote invariance to an unknown shortcut and can result in better finite-sample efficiency compared to commonly used regularization schemes. We empirically validate our theoretical findings by showing that TIPMI and MBM outperform several state-of-the-art methods on one language and two vision datasets.

## 1 Introduction

Neural networks are often deployed on data that is different from the training data — a phenomenon known as distribution shift (Wiles et al., 2021; Cui & Athey, 2022). Many predictors have been shown to have brittle performance under distribution shifts (Kirichenko et al., 2023; Engstrom et al., 2019; Ilyas et al., 2019; Su et al., 2019). One reason for this is shortcut learning: when a predictor relies on easy-to-learn but inconsistent features in the training data that are spuriously correlated with the target label (Geirhos et al., 2020; Makar et al., 2022). Consequently, if the correlation between the shortcut and the main label changes, the model's performance declines significantly. Therefore, for a model to be robust to distribution shifts, it should only rely on features that are both predictive of the label and invariant across the different distributions (Lu et al., 2021).

As a motivating example, consider a predictor trained using X-ray images from a hospital to predict if a patient has knee osteoarthritis (KOA). An ideal predictor would only leverage medically relevant features that are invariant across settings, such as the appearance of the joints or joint space narrowing. However, X-rays often contain inconsistent spurious information that models exploit to make a prediction, such as hospital-specific X-ray artifacts (Zech et al., 2018).

To create models that exclusively learn invariant features, previous methods leverage additional information available only at training time, known as *privileged information*. Most of these approaches utilize *privileged shortcut information*, typically in the form of labels representing potential shortcuts or environments that models should be invariant to (Arjovsky et al., 2019; Makar et al., 2022; Veitch et al., 2021; Sagawa et al., 2019; Goel et al., 2020). Although effective in specific settings, these methods have two main limitations: **(1)** They assume knowledge of all potential shortcuts and the availability of shortcut labels at training time. This is limiting because it requires insight into spurious correlations inherent to a specific training dataset (Schrouff et al., 2024). **(2)** They also rely on invariance penalties that are statistically inefficient. Specifically, they typically rely on nonparametric estimates of distances between distributions that are unreliable especially in contexts where minibatches are used in the training process (Reddi et al., 2015; Ramdas et al., 2015).

In this paper, we address these two limitations by exploring two causally-motivated methods that replace the assumption of known shortcuts with the assumption of access to *privileged mediation information* – intermediate variables that lie on the causal path between input features and target labels, which are available only at training time. We first adapt concept bottleneck models (CBMs) (Koh et al., 2020) to incorporate mediators resulting in a straightforward extension we call Mediator Bottleneck Models (MBMs). Unlike CBM, which initially maps the non-privileged features to low-dimensional, intervenable high-level concepts before making a final prediction, MBM first predicts potentially unstructured, high-dimensional mediators that are then used to make a prediction. However, MBM requires two models for inference, which can be computationally and statistically inefficient if the mediator is high-dimensional or complex. To address this challenge, we propose another method, called Teaching Invariance using Privileged Mediation Information (TIPMI). To avoid the need for a large model at inference time to predict mediators, TIPMI leverages knowledge distillation to enforce invariance to shortcut features. TIPMI first trains a teacher using mediators, followed by a distillation step to train a student that uses normal features.

Our methods address the first limitation of known shortcuts by replacing it with the assumption of known mediators, which is more suitable when expert knowledge of the causal mechanisms linking the label and the input is available. For instance, clinicians know that the effect of KOA on an X-ray is mediated by a narrowing of the knee joint, independent of where the dataset was collected from or the spurious correlations expressed in the data. Moreover, we show that TIPMI and MBM work similarly to the invariance penalties from previous work (Veitch et al., 2021; Makar et al., 2022) while avoiding their inherent inefficiencies.

Our contributions are summarized as follows: **(1)** We study two causally-motivated methods that discourage shortcut learning without requiring *a priori* knowledge of existing shortcuts. The first (MBM) is a simple extension of the well-known CBM while the second (TIPMI) is a novel approach that bypasses the need to deploy multiple models at inference, resulting in lower inference cost. **(2)** We analyze the theoretical properties of the two methods, showing that they lead to more robust models and result in better finite-sample efficiency compared to commonly used regularization schemes. **(3)** We investigate the empirical performance of our methods using one language and two image datasets. Our results show that both methods lead to more robust and efficient models while avoiding the limitations of previous work. The code for our experiments is publicly available at `https://github.com/DylanJamesZapzalka/tipmi-paper`.

## 2   Related Work

**Learning Using Privileged Information and Knowledge Distillation.** Our work builds upon the learning using privileged information (LUPI) paradigm (Vapnik & Vashist, 2009; Lopez-Paz et al., 2015; Lee et al., 2020; Chen et al., 2020; Loquercio et al., 2021; Gao et al., 2019; Chen & Bazzani, 2020), which utilizes privileged information derived from a teacher to train a student. Most LUPI work focuses on gains in efficiency or improved interpretability, in contrast to our main focus: robustness. Related to LUPI is work on knowledge distillation primarily for the purpose of model compression and explainable AI (Buciluǎ et al., 2006; Craven & Shavlik, 1995; Hinton et al., 2015; Menon et al., 2021; Phuong & Lampert, 2019; Yuan et al., 2024). Most similar to our work is (Dao et al., 2020), which views knowledge distillation as a semiparametric inference problem to show how cross-fitting and loss correction can reduce the effects of teacher overfitting and underfitting. Other work has demonstrated that teacher models trained on augmented data can help student models become invariant to known shortcuts (Ojha et al., 2023). In contrast, our work trains the

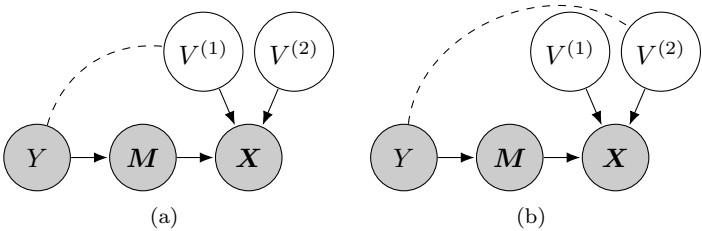

Figure 1: Causal DAGs describing the setting. Grey nodes are observed at training time, while white nodes are never observed. The dashed lines denote non-causal spurious correlations. Each subfigure represents a possible source distribution with the same mediator. In (a) $V^{(1)}$ is a shortcut and in (b) $V^{(2)}$ is a shortcut.

teacher using only mediators, which allows the student to become invariant to unknown shortcuts.

**Shortcut learning.** Previous work on building invariant models explicitly relies on observing a shortcut or environment variable that can be used to induce the desired invariances (Arjovsky et al., 2019; Krueger et al., 2021). Closest to our work is (Makar et al., 2022; Veitch et al., 2021), who present a causally-motivated shortcut removal regularization scheme to encourage robustness to a single shortcut by leveraging observed shortcut variables at training time. While (Zheng & Makar, 2022) encourages invariance to multiple shortcuts, they still require these shortcuts to be observed at training time. As discussed in the introduction, our work avoids the limitations inherent to knowledge or observability of the shortcut/environment label. Discouraging shortcut learning by using data augmentation has also been suggested (Hendrycks et al., 2021; Yin et al., 2019; Cubuk et al., 2018; Wang et al., 2022). This approach can work if we have access to a generator for known shortcut transformations, an assumption that we do not make.

Similar to our work are Concept Bottleneck Models (CBMs) (Koh et al., 2020; Sheth & Ebrahimi Kahou, 2024), which use high-level concepts as privileged information. The concepts are predicted by the first model as intermediate values, and a second model uses the concepts to make a final prediction. Unlike CBMs, TIPMI does not require explicitly modeling the mediator as a function of the main features, allowing for the deployment of just a single small model. This is important in settings where the mediators are high-dimensional. For instance, if the main feature is an image and the mediator is a lengthy text string, a CBM would require a vision language model and a large language model to be deployed at inference time. In contrast, TIPMI would only require a single convolutional neural network for inference.

## 3 Preliminaries

### 3.1 Problem Setting

We consider a supervised learning setting where we wish to learn some model to predict $Y$ from $\boldsymbol{X}$. Additionally, we assume that we have access to privileged mediation information $\boldsymbol{M}$, a variable that fully mediates the causal relationship between $Y$ and $\boldsymbol{X}$. In the KOA example, $\boldsymbol{X}$ and $Y$ represent the X-ray and the presence/severity of KOA, while $\boldsymbol{M}$ could represent measurements of the knee joint. More broadly, in vision applications, segmentations or structured representations of the relevant object or region can serve as mediators that are not influenced by spurious shortcuts, e.g., background artifacts. We assume that $\boldsymbol{M}$ is available only at training time, which happens in settings where data derived from experts may be difficult to acquire at test time due to limited resources. Throughout, we use uppercase letters to denote variables and lowercase to denote their value. Our training data $\mathcal{D} = \{(\boldsymbol{x}_i, \boldsymbol{m}_i, y_i)\}_{i=1}^n$ is drawn from some source distribution $P_s$, where the labels $Y$ are spuriously correlated with $\boldsymbol{X}$ through unknown potential shortcuts $V^{(1)}, V^{(2)}, ..., V^{(d)}$. We denote $\boldsymbol{V}$ as a vector containing all potential shortcuts where $V^{(d)}$ is the $d$'th potential shortcut. Our goal is to create a model $f$ that is invariant to any shortcut represented in $\boldsymbol{V}$.

We assume that the training data is generated in accordance with the causal DAGs in Figure 1, where each DAG represents a possible distribution. For simplicity, we consider a setting where the potential shortcuts are two-dimensional, i.e., $\boldsymbol{V} = [V^{(1)}, V^{(2)}]$. The DAGs represent the anti-causal setting, where $\boldsymbol{X}$ is generated

by $V^{(1)}, V^{(2)}$ and the mediator $\boldsymbol{M}$. Here, $V^{(1)}$ and $V^{(2)}$ are spuriously correlated but not causally related to $Y$, as indicated by the dashed lines in Figure 1. We also assume that $\boldsymbol{M}$ can be fully recovered from $\boldsymbol{X}$. I.e., there exists some unknown deterministic function $\pi$ such that $\boldsymbol{M} := \pi(\boldsymbol{X})$. This assumption follows prior work in robustness and causally motivated methods (Makar et al., 2022), where mediators are modeled as deterministic functions of the inputs to enable clear theoretical analysis. We also note that this assumption is typically satisfied in settings where the input features are rich, unstructured data, such as image or text data.

In the KOA example, shortcuts could be created by spurious correlations between the KOA status and the X-ray image quality. This happens, for example, if patients with more advanced KOA are more likely to receive care in a skilled nursing facility with lower-quality X-rays prone to overexposure $\boldsymbol{V}$, while healthier patients receive care in hospitals with high-quality X-rays. However, we assume that medically relevant information $\boldsymbol{M}$ remains constant across all settings, i.e., narrowing of the knee joint is associated with KOA across all medical facilities. Following this example, we define a family of potential target distributions $\mathcal{P}$, where each target distribution, denoted $P_t$, can be obtained by replacing $P_s(\boldsymbol{V}|Y)$ with some target conditional distribution $P_t(\boldsymbol{V}|Y) \in \mathcal{P}_{V|Y}$:

$$\mathcal{P} = \{P_s(\boldsymbol{X}|\boldsymbol{M}, \boldsymbol{V})P_s(\boldsymbol{M}|Y)P_s(Y)P_t(\boldsymbol{V}|Y) \mid P_t(\boldsymbol{V}|Y) \in \mathcal{P}_{V|Y}\}. \tag{1}$$

Equation 1 highlights the stability of $P_s(\boldsymbol{M})$ across all distributions in $\mathcal{P}$, a desirable property that we will use to inform our approach. This stems from the fact that $Y$ is the only causal parent of $\boldsymbol{M}$, and, crucially, that $\boldsymbol{V}$ does not influence $\boldsymbol{M}$. As a result, $P_t(\boldsymbol{M}) = P_s(\boldsymbol{M})$ for all $P_t \in \mathcal{P}$.

We denote the unconfounded distribution as $P^\circ \in \mathcal{P}$, where $P^\circ = P_s(\boldsymbol{X}|\boldsymbol{M}, \boldsymbol{V})P_s(\boldsymbol{M}|Y)P_s(Y)P^\circ(\boldsymbol{V})$ (Makar et al., 2022; Feder et al., 2024). Unless $P_s = P^\circ$, the Bayes optimal predictor will not be invariant to spurious correlations between $\boldsymbol{X}$ and $Y$ (Makar et al., 2022; Veitch et al., 2021). This is because there are two open pathways between $\boldsymbol{X}$ and $Y$: the front door path, $\boldsymbol{X} \to \boldsymbol{M} \to Y$, and the back door path, $\boldsymbol{X} \to \boldsymbol{V} \to Y$. Our approach hinges on using $\boldsymbol{M}$ at training to discourage the use of the back door path. We note our setup is similar to the front-door criterion used in causal effect estimation (Bellemare et al., 2024), as both rely on mediators to separate spurious from causal associations. Unlike the front-door setting, we focus on anti-causal prediction, aiming for models invariant to shortcuts rather than causal effect estimation.

### 3.2 Counterfactual Invariance

Using potential outcomes notation (Rubin, 2005), we denote $\boldsymbol{X}(\boldsymbol{v})$ as the counterfactual value of $\boldsymbol{X}$ where some $\boldsymbol{V}$ is set to $\boldsymbol{v}$, and all other variables remain the same. Since the association between $\boldsymbol{V}$ and $Y$ is purely spurious, the predictions made by a robust model should not change if the input to the model is $\boldsymbol{X}(\boldsymbol{v})$ or $\boldsymbol{X}(\boldsymbol{v}')$, where $\boldsymbol{v} \neq \boldsymbol{v}'$. We say that models that satisfy this robustness property are counterfactually invariant. This definition extends the definition from (Veitch et al., 2021) to account for all potential shortcuts.

**Definition 1.** *Let $\boldsymbol{V} = [V^{(1)}, V^{(2)}, ..., V^{(d)}]$ contain all possible shortcuts. A model $f(\boldsymbol{X})$ is counterfactually invariant if $f(\boldsymbol{X}(\boldsymbol{v})) = f(\boldsymbol{X}(\boldsymbol{v}'))$ for all $\boldsymbol{v}$, $\boldsymbol{v}'$ in the sample space of $\boldsymbol{V}$.*

Under the assumptions encoded in the DAGs of Figure 1, it follows that a model is counterfactually invariant if it only relies on the mediator $\boldsymbol{M}$. We state this formally in Lemma 1, which is restated from Lemma 3.1 in Veitch et al. (2021).

**Lemma 1.** *(Restated Lemma 3 From Veitch et al. (2021)) Let $\boldsymbol{M}$ be a $\boldsymbol{X}$-measurable random variable such that, for all measurable functions $f$, we have that $f$ is counterfactually invariant if and only if $f(\boldsymbol{X})$ is $\boldsymbol{M}$-measurable.*

Intuitively, this follows as $\boldsymbol{M}$ is unaffected by any intervention on $\boldsymbol{V}$. Lemma 1 motivates our approach for building invariant models by leveraging information encoded in $\boldsymbol{M}$.

## 4 Enforcing invariance using mediators

Recall that our primary goal is to train a model that is *accurate* and *robust* to distribution shifts across all $P_t \in \mathcal{P}$. Here, we describe our approaches for achieving this goal.

### 4.1 An inefficient solution: invariance through conditional independence

Lemma 1 is helpful in identifying that the information encoded in $\boldsymbol{M}$ is unique in that it gives us an "invariance signature." However, it does not specify *how* to use $\boldsymbol{M}$ to generate counterfactually invariant predictions since $\boldsymbol{M}$ is not available at test time. Similar to Veitch et al. (2021), one way to achieve our goal would be to train a model $f$ that has the same conditional independence as the unknown counterfactual invariant predictor, e.g., $f(\boldsymbol{X}) \perp\!\!\!\perp Y|\boldsymbol{M}$. This solution has two advantages: (1) it enforces invariance to shortcuts without requiring access to $\boldsymbol{M}$ at test time, and (2) it does not require knowledge of the shortcuts. However, this approach inherits the second limitation of existing work, especially when $\boldsymbol{M}$ is high dimensional. In those settings, enforcing this signature requires a large sample size, which is inconsistent with settings where training is done in minibatches (Ramdas et al., 2015; Reddi et al., 2015).

Instead, we propose two other approaches that address the limitations of existing work. Both approaches leverage $\boldsymbol{M}$ to create counterfactually invariant models without requiring *a priori* knowledge of possible shortcuts *and* without relying on statistically inefficient conditional invariance tests.

### 4.2 Mediator Bottleneck Models

Based on Lemma 1, a model that is only a function of $\boldsymbol{M}$ should be counterfactually invariant. Therefore, a valid strategy is to learn a representation of $\boldsymbol{X}$ that only captures information relevant to $\boldsymbol{M}$, which can be done by training a model to predict $\boldsymbol{M}$ from $\boldsymbol{X}$. Since we assume a deterministic mapping between $\boldsymbol{X}$ and $\boldsymbol{M}$ exists, a model will not need to rely on shortcuts to reliably predict $\boldsymbol{M}$. The prediction from this first model provides us with a valid, shortcut-free representation that we can then use to predict the target label. We call this approach the Mediator Bottleneck Model (MBM), an adaptation of CBM in which a mediator model $f^M : \boldsymbol{X} \to \boldsymbol{M}$ maps non-privileged features to mediator variables, and a target model $g^M : \boldsymbol{M} \to Y$ maps mediators to the final prediction, i.e., given $\boldsymbol{x} \in \mathcal{X}$, prediction at inference proceeds as $g^M \circ f^M(\boldsymbol{x})$. For the mediator and target model function classes $\mathcal{F}_M$ and $\mathcal{G}_M$, the MBM function class is defined as

$$\mathcal{F}_{\mathrm{MBM}}(f^M) = \{g^M \circ f^M \,|\, g^M \in \mathcal{G}_M\}.$$

In practice, we train MBMs sequentially, similar to CBMs as described in Koh et al. (2020): we first train the mediator model $f^M$ and then train the target model $g^M$ using the mediator predictions of $f^M$ using the dataset $\mathcal{D}$. While this training pipeline mirrors that of CBMs, there are both practical and conceptual distinctions. Notably, MBMs do not assume that intermediate variables are interpretable or directly intervenable. For example, in the KOA prediction task, CBM would require a low-dimensional set of features, such as an indicator for joint space narrowing or bone spurs. By contrast, MBM can allow the intermediate (mediating) variables to be a segmentation of the knee X-ray, radiologist annotations, or measurements of all medically relevant anotomical structures in the knee. This relaxation allows MBMs to leverage high-dimensional, unstructured mediating variables, in contrast to the low-dimensional, structured concepts typically used in CBMs. While this flexibility broadens the applicability of the bottleneck framework, it also introduces additional computational challenges. We address these in the next section via a novel approach designed to efficiently handle such high-dimensional mediators during training and inference.

### 4.3 Teaching Invariance Using Privileged Mediation Information

We propose an approach that addresses the two limitations of work that relies on access to the shortcut label and provides an alternative to MBM that eschews the computational burden imposed by learning a mapping to a high dimensional mediator. Unlike MBM, TIPMI only requires one model at inference time, and does not involve explicitly modeling the potentially complex mapping from $\boldsymbol{X}$ to $\boldsymbol{M}$.

TIPMI proceeds as follows. First, we train a teacher model $g^T(\boldsymbol{M})$ that predicts the target label $Y$ using the mediators $\boldsymbol{M}$. By Lemma 1, $g^T(\boldsymbol{M})$ should be a counterfactually invariant teacher. Since $\boldsymbol{M}$ is not observed at test time, we use distillation to train a student $f^T(\boldsymbol{X})$ that mimics the behavior of the teacher $g^T(\boldsymbol{M})$ but uses only non-privileged features. This allows the student to adopt the same desirable conditional independences as the teacher without the statistical inefficiency caused by explicitly enforcing a conditional independence penalty. Furthermore, unlike an MBM model, TIPMI does not require learning a mapping

from a high-dimensional input $\boldsymbol{X}$ to a high dimensional label $\boldsymbol{M}$ as a sub-routine, and it only requires a single model to be deployed at inference time. For a the student and teacher model function $\mathcal{F}_T$ and $\mathcal{G}_T$ and a teacher model $g^T(\boldsymbol{M})$, we define the TIMPI function class as:

$$\mathcal{F}_{\text{TIPMI}}(g^T) := \arg\min_{f' \in \mathcal{F}_T} \frac{1}{n} \sum_{i=1}^{n} (f'(\boldsymbol{x}_i) - g^T(\boldsymbol{m}_i))^2.$$

In settings where $\mathcal{G}_T$ is highly expressive (e.g., when using deep neural networks), TIPMI might be prone to overfitting bias. This bias arises when data points are used to fit both the teacher model (a nuisance model) and the student model (the target model). When overfitting occurs, which significantly reduces distillation's benefits under TIPMI. This is because if the teacher produces predictions that are nearly identical to the ground-truth labels $Y$, the student effectively learns to map $X \to Y$ directly, rather than inheriting the invariance properties of the teacher. This makes it similarly susceptible to shortcuts as a standard training setup that maps $X \to Y$. Inspired by semiparametric inference literature, we address this overfitting problem by employing sample splitting. Specifically, we split the data into random, mutually exclusive subsets. We use one subset to train the teacher to predict $Y$ from $\boldsymbol{M}$ and use the mutually exclusive subsets to train the student to match the teacher's predictions from $\boldsymbol{X}$. To avoid losing power due to sample-splitting, we apply cross-fitting, which repeats the above process for $K$-folds. The TIPMI algorithm is given below.

**Teacher (Step 1)** Train multiple teachers using cross-fitting:

1. Give $N$ samples, form $K$ disjoint index sets $I_1, I_2, ..., I_k$ partitioning $\{1, ..., N\}$, each with $|I_k| = \frac{N}{K}$.

2. For each fold $k \in \{1, ..., K\}$ and a suitable loss function $\ell_g$, train a teacher model on the combined folds $\mathcal{D}_k^c = \{(\boldsymbol{m}_i, y_i)\}_{i \in I_k^c}$ where $I_k^c := \{1, 2, ..., N\} \backslash I_k$ as follows

$$\hat{g}_k^T = \arg\min_{g^T} \frac{1}{|\mathcal{D}_k^c|} \sum_{i=1}^{|\mathcal{D}_k^c|} \ell_g(g^T(\boldsymbol{m}_i), y_i)$$

3. For $k \in \{1, 2, ..., K\}$, use the teacher models to generate new datasets where $\hat{y}_i = \hat{g}_k^T(\boldsymbol{m}_i)$:

$$\mathcal{D}_k = \{(\boldsymbol{x}_i, \hat{y}_i)\}_{i \in I_k}$$

4. Combine the datasets generated by each teacher to create the final dataset for the student:

$$\mathcal{D}_{cf} = \mathcal{D}_1 \cup \mathcal{D}_2 \cup ... \cup \mathcal{D}_k$$

**Student (Step 2)** Using the cross-fitting dataset $\mathcal{D}_{cf}$, train the student using the mean squared error (MSE) so that the students predictions match the teachers:

$$\hat{f}^T = \arg\min_{f^T} \frac{1}{n} \sum_{i=1}^{n} (f^T(\boldsymbol{x}_i) - \hat{y}_i)^2. \tag{2}$$

## 5 Theory

In this section, we analyze the theoretical properties of MBM and TIPMI to highlight their robustness (section 5.1) as well as generalization properties (section 5.2).

### 5.1 TIPMI and MBM are robust

We start our analysis of the robustness of TIPMI and MBM by studying their population-level properties. We know that the Bayes optimal classifier for the teacher $g_0^T(\boldsymbol{M}) = \mathbb{E}_{P^s}[Y|\boldsymbol{M}]$ and Bayes optimal mediator model $f_0^{sM}(\boldsymbol{X}) = \mathbb{E}_{P^s}[\boldsymbol{M}|\boldsymbol{X}]$, are the same for any source distribution $P^s$ as implied by Figure 1 and the assumption that $\boldsymbol{M}$ is a deterministic function of $\boldsymbol{X}$. For TIPMI, the distillation process can be viewed as enforcing the signature $f^T(\boldsymbol{X}) = g_0^T(\boldsymbol{M})$. By "mimicking" the teacher $g_0^T(\boldsymbol{M})$, the student, $f^T(\boldsymbol{X})$, inherits the teacher's counterfactual invariance. For the MBM, the target model is counterfactually invariant as it can only rely on $\boldsymbol{M}$ to make predictions. We show these two properties in the next proposition.

**Proposition 1.** *Let $P_s$ be any source distribution defined under the causal DAG in Figure 1. Also, let $g_0^T(M) = \mathbb{E}_{P_s}[Y|M]$, $f_0^T(X) = \mathbb{E}_{P_s}[g_0^T(M)|X]$, $f_0^M(M) = \mathbb{E}_{P_s}[M|X]$, and $g_0^M(X) = \mathbb{E}_{P_s}[Y|f_0^M(X)]$. Then both $f_0^T(X)$ and $g_0^M(X)$ are (i) counterfactually invariant, and (ii) optimal under $P^\circ$.*

See Appendix A for the proof for Proposition 1. This proof follows immediately from the d-separation properties of the causal DAG in Figure 1 and the assumption of recoverability, i.e., $M := \pi(X)$. Specifically, since $M$ can be fully recovered from $X$, the Bayes optimal target model will be a function of only $M$ and the Bayes optimal student can perfectly mimic the Bayes optimal teacher. Without further assumptions, a counterfactually invariant predictor cannot be optimal under any distribution other than $P^\circ$.

## 5.2 Generalization properties of TIPMI and MBM

Having established their robustness, we now turn to an analysis of the generalization error of TIPMI and MBM when trained using data from any source distribution $P^s$ and applied to any target distribution $P^t \in \mathcal{P}$. Our analysis follows arguments similar to those presented by Makar et al. (2022). For simplicity, our theoretical analysis focuses on a setting with a single unknown binary shortcut $V$ and a binary label $Y$, even though our proposed approach naturally extends to settings with multiple shortcuts. We leave formal analysis of multiple unknown shortcuts for future work. We focus on finding an upper bound for the generalization error between the risk over a target distribution and the empirical risk over the source distribution, which can be decomposed as follows:

$$R_{P^t} - \hat{R}_{P^s} = (R_{P^t} - R_{P^s}) + (R_{P^s} - \hat{R}_{P^s}),$$

where $R_{P^t}$ is the population risk over $P^t \in \mathcal{P}$, with $R_{P^t} = \mathbb{E}_{X,Y \sim P^t}[\ell(f(X), Y)]$ and $\hat{R}_{P^s} = \frac{1}{n}\sum_{i=1}^n \ell(f(x_i), y_i))$, is the empirical risk for some loss function $\ell$. The first term, $R_{P^t} - R_{P^s}$, which we refer to as the structural risk gap, measures the difference in population risk between some target distribution $P^t$ and the source distribution $P^s$ used to train a model $f$. The second term, $R_{P^s} - \hat{R}_{P^s}$, which we refer to as the learning gap, measures how well the model can generalize in-distribution. In the remainder of this section, we will show that the TIPMI and MBM objectives can control both terms.

### 5.2.1 Controlling the structural risk gap

Here, we study the structural risk gap, $R_{P^t} - R_{P^s}$. Note that this term does not include any estimation error. It is simply the population-level difference between the risks when testing under different disributions.

**Proposition 2.** *Let $\Omega$ be the unit ball in a reproducing kernel Hilbert space with an L-Lipchitz feature mapping used in the MMD. Also let the $g^M$ be K-Lipchitz, $P^s$ be some distribution that conforms to the causal DAGs in Figure 1, and $P^t \in \mathcal{P}$. Then*

$$\text{For TIPMI:} \quad |R_{P^t}(f^T) - R_{P^s}(f^T)| \leq 4L \sup_{P_{vy}^s} \mathbb{E}_{P_{vy}^s}[|g^T(M) - f^T(X)|]$$

$$\text{For MBM:} \quad |R_{P^t}(g^M \circ f^M) - R_{P^s}(g^M \circ f^M)| \leq 4LK \sup_{P_{vy}^s} \mathbb{E}_{P_{vy}^s}[\|f^M(X) - M\|]$$

See Appendix A for the proof for Proposition 2. For TIPMI, Proposition 2 shows that by encouraging the student predictor $f^T$ to be as close as possible to the teacher $g^T$, the structural risk gap becomes smaller. As for MBM, as long as the mediator model $f^M(X)$ is able to accurately predict $M$, the full model $g^M \circ f^M$ will have a low structural risk gap.

### 5.2.2 Controlling the learning gap

Next, we shift our focus to studying the learning gap, $R_{P^s} - \hat{R}_{P^s}$. For our analysis, we compare the Rademacher complexities of the TIPMI student, MBM model, and normal L2 regularization function classes. We also show that TIPMI and MBM can lead to models with better finite-sample efficiency compared to L2 regularization. To simplify our exposition, we consider the special case of linear binary classifiers, where the models take the form of $h(x) = \sigma(w^T x)$ and $\sigma(x)$ denotes the sigmoid function. In addition, we assume that each model has L2 regularization, such that $\|w\|_2 \leq A$, e.g., $\mathcal{F}_{L2} = \mathcal{G}_M = \mathcal{F}_T$, where

$\mathcal{F}_{\text{L2}} := \{f : \boldsymbol{x} \to \sigma(\boldsymbol{w}^T\boldsymbol{x}), \|\boldsymbol{w}\|_2 \leq A\}$. Extensions of our analysis to deeper neural networks can be done using tools presented in (Golowich et al., 2018). We define the vector $\Delta$ that represents the average change in $\boldsymbol{X}$ given some intervention on any unknown potential shortcut $V$. Specifically, $\Delta$ is defined as follows:

$$\Delta := \mathbb{E}_{P^s}[\boldsymbol{X}|\text{do}(V=1)] - \mathbb{E}_{P^s}[\boldsymbol{X}|\text{do}(V=0)]$$
$$= P(Y=1)(\mathbb{E}[\boldsymbol{X}|V=1,Y=1] - \mathbb{E}[\boldsymbol{X}|V=0,Y=1])$$
$$+ P(Y=0)(\mathbb{E}[\boldsymbol{X}|V=1,Y=0] - \mathbb{E}[\boldsymbol{X}|V=0,Y=0])$$

Next, we define the projection matrix $\Pi := \Delta(\Delta^\top\Delta)^{-1}\Delta^\top$, which will project any vector onto the shortcut subspace, whereas $(I - \Pi)$ will project any vector onto the invariant mediator subspace where $\boldsymbol{m}_\perp := \Pi\boldsymbol{x}$ and $\boldsymbol{m}_\parallel := (I - \Pi)\boldsymbol{x}$. This is as $\boldsymbol{X}$ is only a function of $\boldsymbol{M}$ and V when there is a single shortcut, as implied by the DAGs in Figure 1. Here, we can think of $\boldsymbol{m}_\perp$ as the part of $\boldsymbol{x}$ that contains all irrelevant or spurious information and $\boldsymbol{m}_\parallel$ as the part of $\boldsymbol{x}$ that contains mediator information. Similarly, we define $\boldsymbol{w}_\perp := \Pi\boldsymbol{w}$ and $\boldsymbol{w}_\parallel := (I - \Pi)\boldsymbol{w}$. Our next result shows that TIPMI reduces both the teacher and student's error by controlling the influence of variations orthogonal to the mediator.

**Proposition 3.** *Let* $\boldsymbol{m}_\perp := \Pi\boldsymbol{x}$ *and* $\boldsymbol{m}_\parallel := (I - \Pi)\boldsymbol{x}$, *and* $\mathcal{R}(\mathcal{F})$ *be the Rademacher complexity of some function space* $\mathcal{F}$. *For training data* $\mathcal{D} = \{(\boldsymbol{x}_i, \boldsymbol{m}_i, y_i)\}_{i=1}^n$ *where* $\mathcal{D} \sim P^s$, *also we have that* $\sup_{\boldsymbol{m}_\perp} \|\boldsymbol{m}_\perp\|_2 \leq B_\perp$, *and* $\sup_{\boldsymbol{m}_\parallel} \|\boldsymbol{m}_\parallel\|_2 \leq B_\parallel$. *Finally, let* $4L\sup_{P_{vy}^s} \mathbb{E}_{P_{vy}^s}[\|g^T(\boldsymbol{M}) - f^T(\boldsymbol{X})\|] \leq \tau$ *and* $4LK\sup_{P_{vy}^s} \mathbb{E}_{P_{vy}^s}[\|f^M(\boldsymbol{X}) - \boldsymbol{M}\|] \leq \tau'$. *Then*

$$\mathcal{R}(\mathcal{F}_{L2}) \leq \frac{A\sqrt{B_\parallel^2 + B_\perp^2}}{\sqrt{n}} \quad , \quad \mathcal{R}(\mathcal{F}_{TIPMI}) \leq \frac{A \cdot B_\parallel + \tau\frac{B_\perp}{\|\Delta\|}}{\sqrt{n}} \quad and \quad \mathcal{R}(\mathcal{F}_{MBM}) \leq \frac{A \cdot B_\parallel + \tau'\frac{B_\perp}{\|\Delta\|}}{\sqrt{n}}$$

Proposition 3 shows that the upper bound on $\mathcal{F}_{\text{TIPMI}}$ is tighter than that on $\mathcal{F}_{\text{L2}}$ when the student effectively matches the teacher's predictions through knowledge distillation. For MBM, the same Proposition indicates that the upper bound on $\mathcal{F}_{\text{MBM}}$ is lower than that the on $\mathcal{F}_{\text{L2}}$ when the mediator model generalizes well across all groups of $P^s$. Moreover, the bound on $\mathcal{F}_{\text{TIPMI}}$ will likely be tighter than that on $\mathcal{F}_{\text{MBM}}$ when the mapping from $\mathcal{X}$ to $\mathcal{M}$ is very complex. It is important to note, however, that unless $P^s = P^\circ$, the reduction in the student and MBM Rademacher complexity do not necessarily result in improved in-distribution accuracy. This is because both TIPMI and MBM are limited in their ability to leverage shortcut features, i.e., they will exhibit bias unless $P^s = P^\circ$, as they are limited in their use of the shortcuts to predict the label.

## 6 Experiments

We evaluate the proposed models, TIPMI and MBM, across diverse settings to assess their robustness to spurious correlations and sample efficiency. We structure our analysis around three main questions: (1) Do TIPMI and MBM remain robust in the presence of unknown or unobserved shortcuts? (2) How do they compare to baselines in terms of finite-sample efficiency? (3) When is TIPMI preferable to MBM?

### 6.1 Setup

**Datasets.** We evaluate both methods on three datasets, including vision and language tasks:

*Waterbirds.* A binary image classification task where the target label is the type of bird (land versus water birds) (Sagawa et al., 2019). Shortcut labels spuriously correlated with the type of bird are background type $(V^{(1)})$ and image artifacts $(V^{(2)})$. The mediator $\boldsymbol{M}$ is a segmentation mask of the bird foreground.

*Knee osteoarthritis.* A binary classification task using knee X-rays to predict the presence of osteoarthritis. The label is spuriously correlated with rectangular artifacts $(V^{(1)}, V^{(2)})$ mimicking radiographic markers (Zech et al., 2018). The mediator $\boldsymbol{M}$ consists of 16 joint space width measurements.

*Food Review.* A text classification task constructed from the Amazon Food Reviews dataset (McAuley & Leskovec, 2013). Inputs are review texts $(\boldsymbol{X})$, labels are star ratings $(Y)$, and review summaries serve as the mediator $(\boldsymbol{M})$. Both shortcuts $(V^{(1)}, V^{(2)})$ are generated by perturbing high-frequency function words following Veitch et al. (2021), e.g., replacing "a" with "axxxxx" for $V^{(1)}$ and "to" with "toyyyyy" for $V^{(2)}$.

**Baselines.** We evaluate TIPMI and MBM against a range of established baselines, as well as ablation variants designed to isolate the effects of cross-fitting: (1) **Mediator-Conditional Penalization (MCPM):** Enforces $f(\boldsymbol{X}) \perp Y \mid \boldsymbol{M}$ during training via the kernel conditional independence test (KCIT) (Zhang et al., 2012). (2) **Shortcut-Conditional Penalization (SCPM):** Has oracle access to a (possibly incomplete) set of shortcuts $\boldsymbol{V}$ and enforces $f(\boldsymbol{X}) \perp \boldsymbol{V} \mid Y$ using kernel-based MMD (Gretton et al., 2012; Veitch et al., 2021; Makar et al., 2022). (3) **Group Distributionally Robust Optimization (GDRO):** Has oracle access to a (possibly incomplete) set of shortcuts $\boldsymbol{V}$ and minimizes the worst-group loss, where the groups are defined by shortcut and label values (Sagawa et al., 2019). (4) **Invariant Risk Minimization (IRM):** Has oracle access to a (possibly incomplete) set of shortcuts $\boldsymbol{V}$ and learns predictors invariant across environments defined by shortcut and label values (Arjovsky et al., 2019). (5) **Self-Distillation (SD):** A teacher-student setup with identical architectures. The teacher is trained to predict the target label while the student is trained to match the teacher's predictions. (6) **L2 Regularization (L2):** Standard $\ell_2$-penalized empirical risk minimization. (7) **TIPMI-NC:** TIPMI without cross-fitting and **MBM-CF:** MBM with cross-fitting.

**Experiment setup.** In each experiment, we simulate spurious correlations at training time and evaluate the performance on different out-of-distribution shifts where the correlation between the label and shortcut varies. Performance is measured via the area under the ROC curve (AUROC). Models invariant to shortcuts should exhibit stable AUROC across test distributions, while shortcut-reliant models will degrade under distribution shift. Each experiment is repeated with 10 random seeds; error bars denote standard error.

For the waterbirds experiments, the TIPMI teacher, student, and MBM target models are ResNet-50 models (He et al., 2016) pretrained on ImageNet-1k (Russakovsky et al., 2015), while the MBM mediator model is a U-Net with a ResNet-50 encoder. For KOA, the TIPMI student and MBM mediator models are ResNet-50 models, and the TIPMI teacher and MBM target models are single-layer neural networks with 1024 hidden units. For the food review experiments, the TIPMI teacher, student, and MBM target models are BERT-tiny classifiers (Bhargava et al., 2021), and the MBM mediator model is a T5-small model (Raffel et al., 2020). Additional training details are provided in the Appendix.

## 6.2 Robustness to unknown/unobserved shortcuts

We evaluate the robustness in settings with two spurious features, where only one is known/observed. This setting highlights the limitations of building robustness solely to known shortcuts. To evaluate robustness to unknown shortcuts, we restrict access to shortcut labels during training: methods relying on known shortcuts (SCPM, GDRO, IRM) are given access to only $V^{(1)}$ but not $V^{(2)}$. Spurious correlations are introduced during training by setting $P_s(V^{(d)} = 1 | Y = 1) = P_s(V^{(d)} = 0 | Y = 0) = 0.95$ in the waterbirds dataset and $P_s(V^{(d)} = 1 | Y = 1) = P_s(V^{(d)} = 0 | Y = 0) = 0.9$ in the food review and KOA datasets, for each shortcut $V^{(1)}$ and $V^{(2)}$. At test time we fix the distribution over the known shortcut $P_t(V^{(1)} = 1 | Y = 1) = P_t(V^{(1)} = 0 | Y = 0) = 0.5$ and vary $P_t(V^{(2)} | Y)$ to assess sensitivity to the unobserved shortcut.

**Baselines rely on shortcuts.** First, we compare TIPMI and MBM to other baselines in Figure 2. Following the ablation study, we exclude TIPMI-NC and MBM-CF. Across all datasets, TIPMI and MBM exhibit strong robustness to the unobserved shortcut, maintaining high and stable AUROC. In contrast, methods requiring known shortcuts (SCPM, GDRO, and IRM) degrade as the shortcut-label correlation deviates from training. Notably, on KOA and food review, SCPM underperforms even standard L2-regularized ERM, indicating that reliance on partial shortcut information is insufficient for robustness. MCPM also performs worse than TIPMI and MBM, despite using mediators to enforce invariance, highlighting the inefficiency of the conditional independence penalty. On waterbirds and KOA, TIPMI and MBM perform similarly and are more robust than the baselines. MBM performs better than TIPMI in food review, as removing the shortcut when mapping $\boldsymbol{X}$ to $\boldsymbol{M}$ is easy (e.g., summarizing text), since the complexity is offset due to large amounts of pretraining data used by the mediator model (e.g., T5 model). SD results are shown only in the appendix, as they mirror L2, and IRM is excluded due to poor performance on waterbirds and KOA.

**Ablation study.** Next, we perform an ablation study to determine the benefits of cross-fitting for both TIPMI and MBM. Figure 3 shows results for the waterbirds, KOA, and food review datasets. TIPMI-NC in the waterbirds and the food review datasets perform significantly worse than TIPMI, highlighting the need

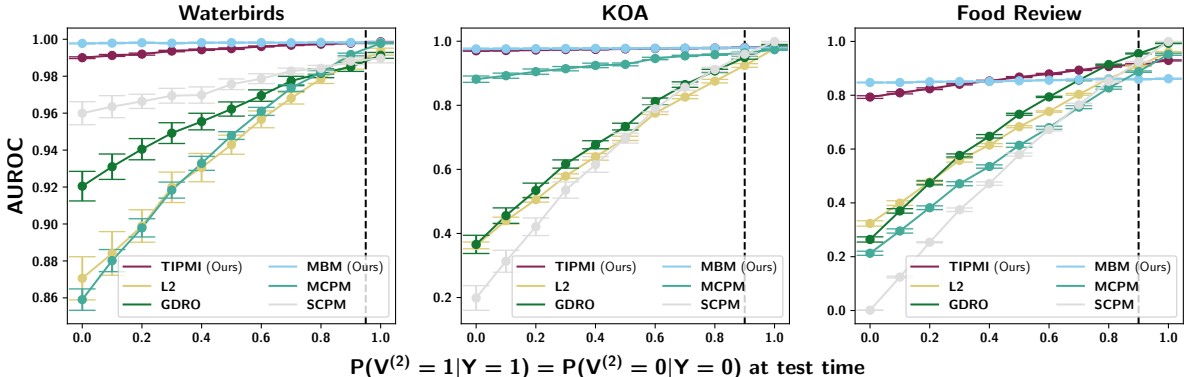

Figure 2: $x$-axis shows $P(V^{(2)}|Y)$ at test time. $y$-axis shows the AUROC over the test data. Dashed vertical line shows $P(V^{(2)}|Y)$ at training time. **(Left)** waterbirds and **(Middle)** KOA: MBM and TIPMI outperform all baselines. **(Right)** food review: MBM is the most robust to shortcuts, followed closely by TIPMI.

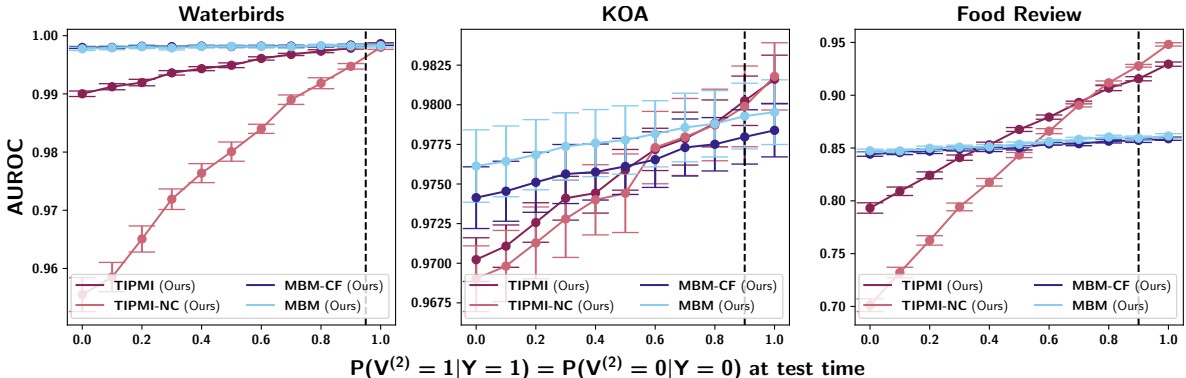

Figure 3: $x$-axis shows $P(V^{(2)}|Y)$ at test time. $y$-axis shows the AUROC over the test data. Dashed vertical line shows $P(V^{(2)}|Y)$ at training time. **(Left)** waterbirds and **(Right)** food review: cross-fitting significantly improves TIPMI, but not MBM. **(Middle)** KOA: cross-fitting has no significant impact on TIPMI or MBM.

for cross-fitting when the teacher is a large and flexible model, such as BERT and ResNet-50 (He et al., 2016; Devlin, 2018). In contrast, there is no difference between TIPMI and TIPMI-NC for the KOA dataset. This is expected as the teacher used in these experiments is a small, single-layer neural network, which is less prone to overfitting than the large networks used in the other experiments. In contrast to TIPMI, cross-fitting has little effect on the robustness of MBM. As shown in Proposition 2, this is because the robustness of MBM is only determined by how well the mediator model can map $\boldsymbol{X}$ to $\boldsymbol{M}$, which is not helped by cross-fitting.

### 6.3 Robustness and accuracy due to finite-sample efficiency

We next highlight how the efficiency of our approaches leads to improved robustness and accuracy.

**Efficiency leads to better robustness.** We first examine how efficiently enforcing counterfactual invariance improves robustness. We study a setting similar to that of section 6.2, but here we focus on robustness to a single known shortcut. In this setting, methods that rely on knowledge of the shortcut and our proposed approaches are – in principle – enforcing the same desired invariance with respect to the shortcut. We simulate datasets containing one shortcut $P_s(V^{(1)} = 1|Y = 1) = P_s(V^{(1)} = 0|Y = 0) = 0.95$ in the waterbirds dataset and $P_s(V^{(1)} = 1|Y = 1) = P_s(V^{(1)} = 0|Y = 0) = 0.9$ in the food review and KOA datasets at training. SCPM, GDRO, and IRM have access to $V^{(1)}$ at training. We vary $P(V^{(1)}|Y)$ at test time.

Results are shown in Figure 4. Although SCPM and GDRO are explicitly enforcing invariance to the relevant shortcut, they are less robust than TIPMI and MBM across all datasets. Similar to section 6.2, MCPM

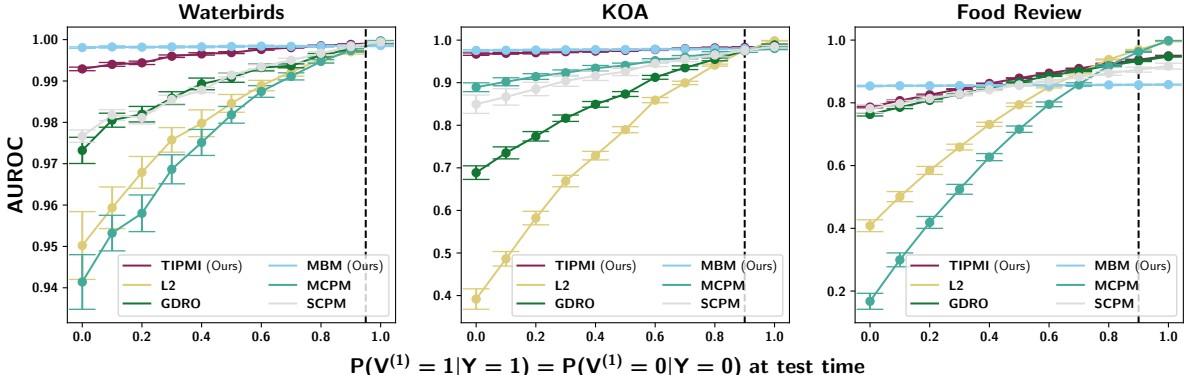

Figure 4: $x$-axis shows $P(V^{(1)}|Y)$ at test time. $y$-axis shows the AUROC over the test data. Dashed vertical line shows $P(V^{(1)}|Y)$ at training time. **(Left)** Waterbirds and **(Middle)** KOA: MBM and TIPMI outperform all baselines. **(Right)** Food review: MBM is most robust, TIPMI is tied for second best.

delivers worse robustness despite enforcing a similar invariance penalty to TIPMI and MBM. MBM performs slightly better than TIPMI, which, at worst, performs comparably to SCPM in the food review task. Here, the superior performance of TIPMI and MBM arises because they are enforcing counterfactual invariance without relying on expensive penalty terms, e.g., MCPM and SCPM rely on data-intensive nonparametric conditional independence penalties to enforce invariance through regularization. MBM performs better than TIPMI in food review and waterbirds, for the same resons discussed in section 6.2.

Table 1: In distribution results.

| Model \ Data | Waterbirds | KOA | Food Review |
|---|---|---|---|
| TIPMI | **0.992 (0.002)** | **0.980 (0.004)** | **0.896 (0.007)** |
| L2 | **0.987 (0.004)** | 0.946 (0.010) | **0.907 (0.006)** |
| GDRO | **0.986 (0.006)** | 0.948 (0.008) | **0.903 (0.007)** |
| MCPM | 0.985 (0.004) | 0.948 (0.008) | **0.899 (0.007)** |
| MBM | **0.993 (0.003)** | **0.980 (0.006)** | 0.857 (0.005) |

**Efficiency leads to better accuracy.** To isolate the gains in finite-sample efficiency introduced by TIPMI and MBM, we run an "in-distribution" experiment where the training and test data are drawn from the unconfounded distribution, $P^\circ$, where $V^{(1)}$ is simulated such that $V^{(1)} \perp\!\!\!\perp Y$. The results, summarized in table 1, show that TIPMI and MBM achieve the best performance on waterbirds and KOA. This is likely because (1) the mediator provides a strong signal for the label, and (2) the mapping $\boldsymbol{X} \to \boldsymbol{M}$ is simplified by pre-training. Specifically, in both waterbirds and KOA, the models are ResNet-50 architectures pre-trained on ImageNet, which offsets the complexity of the mapping. For the food review dataset, MBM is the only model whose performance is statistically distinguishable from the others.

## 6.4 TIPMI vs MBM

Previous results show that MBM and TIPMI are both more robust and efficient than baselines. The results also seem to imply that MBM performs as well as or better than TIPMI. We examine this relationship here.

**MBM has higher inference-time complexity.** First, we highlight that MBM's performance comes at the cost of higher inference-time complexity, as it requires deploying two, potentially large models, whereas TIPMI only requires a single model. Therefore, when the MBM target and TIPMI student model architectures are the same (as for the waterbirds and food review experiments), MBM will require more parameters at inference due to the additional mediator model. And when the mapping between $\boldsymbol{X}$ and $\boldsymbol{M}$ is complex, MBMs parameter count is substantially higher compared to TIPMI. Table 2 shows the number of parameters

Table 2: Number of parameters (in millions) used for TIPMI and MBM at inference time

| Model \ Data | WB | KOA | FR |
|---|---|---|---|
| TIPMI | **23.51** | **23.51** | **4.39** |
| MBM | 56.03 | 23.56 | 64.89 |

for the deployed model (i.e., at inference time) in millions: on the waterbirds (WB) dataset, MBM uses over twice as many parameters as TIPMI, and on the food review (FR) dataset, more than ten times as many.

**MBM performs worse if the mapping from $X$ to $M$ is complex.** This additional complexity comes at the cost of worse finite-sample efficiency, as is shown by the food review results in table 1. This happens as the mediator model is unable to fully recover $M$ from $X$, leaving the target model with a less reliable signal to make a prediction.

We investigate this phenomenon using simulated data where we can control the function complexity. We simulate an anti-causal setting where the complexity of the mapping between $X$ and $M$ can vary. To construct this setting, we designate $D$ dimensions of $X$ that are functions of $M$ but contain no information about the target $Y$. We refer to these as *redundant dimensions*. Increasing $D$ makes the mapping $X \to M$ more complex, while leaving the effective mapping $X \to Y$ unchanged. Our data generation process is:

$$V \sim \text{Normal}(0,1),$$
$$Y^C \sim \text{Normal}(0,1), \ Y^V := \beta_V V_i, \ Y := Y^C + Y^V,$$
$$M^C \sim \beta_Y Y^C + \text{Normal}(0,1), \ \boldsymbol{M}^R \sim \text{Normal}(\boldsymbol{0}, I_D), \ \boldsymbol{M} := [M^C, \boldsymbol{M}^R],$$
$$X^C \sim \beta_M M^C, \ \boldsymbol{X}^R \sim \beta_V V + \sqrt[3]{\boldsymbol{M}^R} + \text{Normal}(\boldsymbol{0}, I_D), \ \boldsymbol{X} := [X^C, \boldsymbol{X}^R],$$

where $Y^C, M^C, X^C$ denote "clean" components of $Y$, $M$, and $X$ unaffected by the shortcut $V$, while $Y^V$ captures the $V$'s contribution to $Y$. The redundant components $\boldsymbol{M}^R$ and $\boldsymbol{X}^R$ are unrelated to the robust component of the target label $Y^C$, with $\boldsymbol{X}^R$ also influenced by $V$. Random weights $\beta_V$, $\beta_M$, and $\beta_Y$ specify mappings from $V \to Y^V$, $M^C \to X^C$, and $Y^C \to M^C$. This setup ensures the data respects the causal DAG: $V$ affects $Y$ and $X$ but not $M$, while decoupling the complexity of $X \to Y$ from that of $X \to M$.

All models are trained with a fifth-degree polynomial kernel and L2 regularization, making the true models attainable (the most complex mapping, MBM's $X \to M$, is a third-degree polynomial). Hyperparameters are chosen via five-fold cross-validation. Each experiment is repeated 100 times with independent draws of $V, \boldsymbol{X}, \boldsymbol{M}, Y$, $\beta_V$, $\beta_Y$, and $\beta_M$. For out-of-distribution evaluation, we flip the shortcut effect by setting $\beta_V := -\beta_V$. Each simulation uses 500 training and 100 test samples. We report the median and interquartile range (IQR) of the root mean squared error (RMSE) across 100 runs. Additional details are in Appendix D.

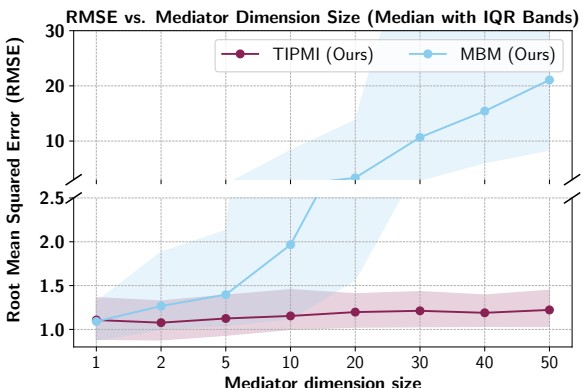

Figure 5: $x$-axis is mediator dimension size. $y$-axis is test RMSE. MBM's performance deteriorates with larger dimensions whereas TIPMI and L2 are stable.

Figure 5 shows MBM's error rises sharply as the number of redundant mediator dimensions increases, while TIPMI's error remains stable. This is because MBM's mediator model is trained to predict all dimensions of $M$, including those irrelevant for predicting $Y$ which introduces additional noise that reduces the quality of the signal for target model to predict the label. TIPMI bypasses this: the student model only needs to learn a mapping from $X$ to the teacher's predictions, without modeling the redundant mediator dimensions.

# 7 Conclusion

In this work, we presented TIPMI and MBM, methods that use privileged mediation information to enforce counterfactual invariance and improve finite-sample efficiency. TIPMI consists of a teacher trained using privileged mediation information and a student that learns through distillation. In contrast, MBM first predicts the mediator before using it to make a final prediction. We showed theoretically and empirically that TIPMI and MBM promote invariance to shortcuts better than methods that use privileged shortcut information and that they increase finite-sample efficiency.

**Limitations and Future Work.** A limitation of our methods is the assumption that shortcuts do not causally affect the mediator. In this case, TIPMI and MBM can still achieve invariance to other shortcuts that don't affect the mediator, but they will not enforce counterfactual invariance. Future work could explore uncertainty analysis to assess performance when part of the mediator is causally affected by a shortcut. Additionally, our theoretical analysis for the generalization properties of TIPMI and MBM is constrained to the setting of a single binary shortcut. Future work could generalize this theory to account for multiple unknown shortcuts.

# 8 Acknowledgements and Disclosure of Funding

We are thankful for the anonymous reviewers and for feedback from Alex D'Amour. This material is based upon work supported by the National Science Foundation under Grants No. 2337529 and 2153083. Any opinions, findings, and conclusions or recommendations expressed in this material are those of the author(s) and do not necessarily reflect the views of the National Science Foundation.

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

## A  Section 5.1 Proofs

For this proof, we show that TIPMI and MBM yield models that are asymptotically counterfactually invariant and optimal under the unconfounded distribution $P^\circ$. This follows directly from the d-separation properties of the DAG in Figure 1 and the assumption that $\boldsymbol{M}$ can be fully recovered from $\boldsymbol{X}$. Without further assumptions, MBM and TIPMI are only optimal under $P^\circ$, as they cannot exploit shortcut features that may be informative in other distributions.

**Proposition A.1** (Restated Proposition 1). *Let $P_s$ be any source distribution defined under the causal DAG in Figure 1. Also, let $g_0^T(\boldsymbol{M}) = \mathbb{E}_{P_s}[Y|\boldsymbol{M}]$, $f_0^T(\boldsymbol{X}) = \mathbb{E}_{P_s}[g_0^T(\boldsymbol{M})|\boldsymbol{X}]$, $f_0^M(\boldsymbol{M}) = \mathbb{E}_{P_s}[\boldsymbol{M}|\boldsymbol{X}]$, and $g_0^M(\boldsymbol{X}) = \mathbb{E}_{P_s}[Y|f_0^M(\boldsymbol{X})]$. Then both $f_0^T(\boldsymbol{X})$ and $g_0^M(\boldsymbol{X})$ are (i) counterfactually invariant, and (ii) optimal under $P^\circ$.*

*Proof.* Let $\boldsymbol{X}, \boldsymbol{M}, V \sim P_s$ for some $P_s$. We first show that the TIPMI student model will be counterfactually invariant. Under the assumption that there exists some deterministic function $\pi$ such that $\boldsymbol{M} = \pi(\mathbf{X})$, we know that all information in $\boldsymbol{M}$ can be recovered from $\boldsymbol{X}$. Therefore, since $g_0^T(\boldsymbol{M}) \perp\!\!\!\perp \boldsymbol{X} \,|\, \boldsymbol{M}$ under $P_s$, $\mathbb{E}_{P_s}[g_0^T(\boldsymbol{M})|\boldsymbol{X}] = \mathbb{E}_{P_s}[g_0^T(\boldsymbol{M})|\boldsymbol{X}, \pi(\boldsymbol{X})] = \mathbb{E}_{P_s}[g_0^T(\boldsymbol{M})|\boldsymbol{M}]$. Thus, $f_0^T(\boldsymbol{X})$ can be written as (with abuse of notation) as $f_0^T(\boldsymbol{M})$. Since $f_0^T$ is only a function of $\boldsymbol{M}$, $f_0^T$ must be counterfactually invariant by Lemma 3.1 in (Veitch et al., 2021). This can be seen as an intervention on $\boldsymbol{V}$ does not affect $\boldsymbol{M}$. Also, $f_0^T$ is optimal under $P^\circ$. Similarly, we can show that the MBM target model will be counterfactually invariant, as $g_0^M(\boldsymbol{X}) = \mathbb{E}_{P_s}[Y|f_0^M(\boldsymbol{X})] = \mathbb{E}_{P_s}[Y|\boldsymbol{M}]$.

Both TIPMI student and MBM target model are also optimal under $P^\circ$. The target model is optimal under $P^\circ$ as $\mathbb{E}_{P_s}[Y|\boldsymbol{M}] = \mathbb{E}_{P^\circ}[Y|\boldsymbol{M}] = \mathbb{E}_{P^\circ}[Y|\boldsymbol{X}]$ since $\boldsymbol{M}$ d-separates $Y$ from $\boldsymbol{X}$. Furthermore, the TIPMI student is optimal under $P^\circ$ as $f_0^T(\boldsymbol{X}) = \mathbb{E}_{P_s}[g_0^T(\boldsymbol{M})|\boldsymbol{X}] = \mathbb{E}_{P_s}[g_0^T(\boldsymbol{M})|\boldsymbol{M}] = g_0^T(\boldsymbol{M})$. $\qquad\square$

## B  Section 5.2.2 Proofs

### B.1  Finite-Sample Results

To understand how TIPMI and MBM induce invariance, we build on prior work that enforces conditional independences satisfied by an unknown counterfactually invariant predictor Veitch et al. (2021), e.g., $f(\boldsymbol{X}) \perp\!\!\!\perp V \,|\, Y$. To enforce this property, they use the maximum mean discrepancy (MMD) Gretton et al. (2012). For $P_{f|vy} := P(f(X)|Y = y, V = v)$, the corresponding regularization term, called CMMD, is defined as

$$\begin{aligned}
\mathrm{CMMD}(P, f) := {} & P(y = 1)\mathrm{MMD}(P_{f|11}, P_{f|01}) \\
& + P(y = 0)\mathrm{MMD}(P_{f|10}, P_{f|00}).
\end{aligned}$$

We first establish in Lemma B.1 that minimizing $\mathrm{CMMD}(P^s, f)$ can reduce the structural risk gap. Lemma B.2 then shows that both TIPMI and MBM can decrease $\mathrm{CMMD}(P^s, f)$. Finally, we apply both Lemma B.1 and Lemma B.2 to show that TIMPI and MBM can reduce the structural risk gap, as shown in Proposition 2.

**Lemma B.1.** *Let $P^s$ be some distribution that conforms to the causal DAG in Figure 1, $P^t \in \mathcal{P}$, and $\ell \in \Omega$, where $\Omega$ is a reproducing kernel Hilbert space used in the MMD. Then for some function $f(\boldsymbol{X})$,*

$$|R_{P^t}(f) - R_{P^s}(f)| < CMMD(P^s, f)$$

*Proof.* The structure of this proof follows that of Proposition A2 from (Makar et al., 2022). For this proof, we define $R_{vy}^t := \mathbb{E}_{P^t}[\ell(f(\boldsymbol{X}), Y)|Y = y, V = v]$ and $R_{vy}^s := \mathbb{E}_{P^s}[\ell(f(\boldsymbol{X}), Y)|Y = y, V = v]$.

For any distribution $P^t \in \mathcal{P}$ we can rewrite the risk as follows:

$$
\begin{aligned}
R_{P^t} &= \mathbb{E}_{P^t}[\ell(f(\boldsymbol{X}), Y)] \\
&= \sum_y \mathbb{E}_{P^t}[\ell(f(\boldsymbol{X}), Y)|Y = y]P^t(Y = y) \\
&= \sum_y \mathbb{E}_{P^t}[\ell(f(\boldsymbol{X}), Y)|Y = y, V = 1]P^t(V = 1|Y = y)P^t(Y = y) \\
&\quad + \mathbb{E}_{P^t}[\ell(f(\boldsymbol{X}), Y)|Y = y, V = 0]P^t(V = 0|Y = y)P^t(Y = y) \\
&= \sum_y R_{1y}^t P^t(V = 1|Y = y)P^t(Y = y) + R_{0y}^t P^t(V = 0|Y = y)P^t(Y = y) \\
&= \sum_y P^t(Y = y)[R_{1y}^t P^t(V = 1|Y = y) + R_{0y}^t P^t(V = 0|Y = y)]
\end{aligned}
$$

Therefore, if we take the difference between both risks, we get the following result:

$$
\begin{aligned}
|R_{P^t} - R_{P^s}| &= |\sum_y P^t(Y = y)[R_{1y}^t P^t(V = 1|Y = y) + R_{0y}^t P^t(V = 0|Y = y)] \\
&\quad - \sum_y P^s(Y = y)[R_{1y}^s P^s(V = 1|Y = y) + R_{0y}^s P^s(V = 0|Y = y)]| \\
&= |\sum_y P^s(Y = y)[R_{1y}^s P^t(V = 1|Y = y) - R_{1y}^s P^s(V = 1|Y = y) \\
&\quad + R_{0y}^s P^t(V = 0|Y = y) - R_{0y}^s P^s(V = 0|Y = y)]| \\
&= |\sum_y P^s(Y = y)[(P^t(V = 1|Y = y) - P^s(V = 1|Y = y))R_{1y}^s \\
&\quad + (P^t(V = 0|Y = y) - P^s(V = 0|Y = y))R_{0y}^s]| \\
&= |\sum_y P^s(Y = y)[(P^t(V = 1|Y = y) - P^s(V = 1|Y = y))R_{1y}^s \\
&\quad + (1 - P^t(V = 1|Y = y) - 1 + P^s(V = 1|Y = y))R_{0y}^s]| \\
&= |\sum_y P^s(Y = y)[(P^t(V = 1|Y = y) - P^s(V = 1|Y = y))R_{1y}^s \\
&\quad - (P^t(V = 1|Y = y) - P^s(V = 1|Y = y))R_{0y}^s]| \\
&= |\sum_y P^s(Y = y)(P^t(V = 1|Y = y) - P^s(V = 1|Y = y))(R_{1y}^s - R_{0y}^s)| \\
&< |\sum_y P^s(Y = y)(R_{1y}^s - R_{0y}^s)| \\
&= |P^s(Y = 1)(R_{11}^s - R_{01}^s) + P^s(Y = 0)(R_{10}^s - R_{00}^s)| \\
&\leq |P^s(Y = 1)\mathrm{MMD}(P_{f|11}^s, P_{f|01}^s) + P^s(Y = 0)\mathrm{MMD}(P_{f|10}^s, P_{f|00}^s)| \\
&= \mathrm{CMMD}(P^s, f)
\end{aligned}
$$

$\square$

**Lemma B.2.** *Let $\Omega$ be the unit ball in a reproducing kernel Hilbert space with an L-Lipchitz feature mapping used in the MMD. Also let the $g^M$ be K-Lipchitz, $P^s$ be some distribution that conforms to the causal DAG in Figure 1, and $P^t \in \mathcal{P}$. Then*

$$
CMMD(P^s, f^T) \leq 4L \sup_{P_{vy}^s} \mathbb{E}_{P_{vy}^s}[|g^T(\boldsymbol{M}) - f^T(\boldsymbol{X})|]
$$

*and*

$$CMMD(P^s, g^M \circ f^M) \leq 4LK \sup_{P_{vy}^s} \mathbb{E}_{P_{vy}^s}[\|f^M(\boldsymbol{X}) - \boldsymbol{M}\|]$$

*Proof.* We begin the proof by showing the first bound for the TIPMI student model. The proof follows from the fact that $\mathrm{MMD}(P_{g^T|11}^s, P_{g^T|01}^s) = \mathrm{MMD}(P_{g^T|10}^s, P_{g^T|00}^s) = 0$, as the teacher $g^T$ is invariant across all distributions.

$$\begin{aligned}
\mathrm{CMMD}(P^s, f^T) &= P(y=1)\mathrm{MMD}(P_{f^T|11}^s, P_{f^T|01}^s) + P^s(y=0)\mathrm{MMD}(P_{f^T|10}^s, P_{f^T|00}^s) \\
&\leq \mathrm{MMD}(P_{f^T|11}^s, P_{f^T|01}^s) + \mathrm{MMD}(P_{f^T|10}^s, P_{f^T|00}^s) \\
&= [\mathrm{MMD}(P_{f^T|11}^s, P_{f^T|01}^s) - \mathrm{MMD}(P_{g^T|11}^s, P_{g^T|01}^s)] \\
&\quad + [\mathrm{MMD}(P_{f^T|10}^s, P_{f^T|00}^s) - \mathrm{MMD}(P_{g^T|10}^s, P_{g^T|00}^s)]
\end{aligned}$$

(The equality holds as $g$ is only a function of $\boldsymbol{M}$, which is invariant across all distributions. We can now obtain the final bounds by the L-Lipchitz properties of the feature mapping and the definition of the MMD.)

$$\begin{aligned}
\mathrm{CMMD}(P^s, f^T) &\leq \sup_{\omega \in \Omega}(\mathbb{E}_{P_{11}^s}[\omega(f^T(\boldsymbol{X}))] - \mathbb{E}_{P_{01}^s}[\omega(f^T(\boldsymbol{X}))] - \mathbb{E}_{P_{11}^s}[\omega(g^T(\boldsymbol{M}))] + \mathbb{E}_{P_{01}^s}[\omega(g^T(\boldsymbol{M}))]) \\
&\quad + \sup_{\omega \in \Omega}(\mathbb{E}_{P_{10}^s}[\omega(f^T(\boldsymbol{X}))] - \mathbb{E}_{P_{00}^s}[\omega(f^T(\boldsymbol{X}))] - \mathbb{E}_{P_{10}^s}[\omega(g^T(\boldsymbol{M}))] + \mathbb{E}_{P_{00}^s}[\omega(g^T(\boldsymbol{M}))]) \\
&\leq \sup_{\omega \in \Omega}(\mathbb{E}_{P_{11}^s}[\omega(f^T(\boldsymbol{X})) - \omega(g^T(\boldsymbol{M}))]) + \sup_{\omega \in \Omega}(\mathbb{E}_{P_{01}^s}[\omega(g^T(\boldsymbol{M})) - \omega(f^T(\boldsymbol{X}))]) \\
&\quad + \sup_{\omega \in \Omega}(\mathbb{E}_{P_{10}^s}[\omega(f^T(\boldsymbol{X})) - \omega(g^T(\boldsymbol{M}))]) + \sup_{\omega \in \Omega}(\mathbb{E}_{P_{00}^s}[\omega(g^T(\boldsymbol{M})) - \omega(f^T(\boldsymbol{X}))]) \\
&= \sup_{\omega \in \Omega}(\langle\omega, \mathbb{E}_{P_{11}^s}[\phi(f^T(\boldsymbol{X})) - \phi(g^T(\boldsymbol{M}))]\rangle) + \sup_{\omega \in \Omega}(\langle\omega, \mathbb{E}_{P_{01}^s}[\phi(g^T(\boldsymbol{M})) - \phi(f^T(\boldsymbol{X}))]\rangle) \\
&\quad + \sup_{\omega \in \Omega}(\langle\omega, \mathbb{E}_{P_{10}^s}[\phi(f^T(\boldsymbol{X})) - \phi(g^T(\boldsymbol{M}))]\rangle) + \sup_{\omega \in \Omega}(\langle\omega, \mathbb{E}_{P_{00}^s}[\phi(g^T(\boldsymbol{M})) - \phi(f^T(\boldsymbol{X}))]\rangle) \\
&\leq \sup_{\omega \in \Omega}(\|\omega\| \, \|\mathbb{E}_{P_{11}^s}[\phi(f^T(\boldsymbol{X})) - \phi(g^T(\boldsymbol{M}))]\|) + \sup_{\omega \in \Omega}(\|\omega\| \, \|\mathbb{E}_{P_{01}^s}[\phi(g^T(\boldsymbol{M})) - \phi(f^T(\boldsymbol{X}))]\|) \\
&\quad + \sup_{\omega \in \Omega}(\|\omega\| \, \|\mathbb{E}_{P_{10}^s}[\phi(f^T(\boldsymbol{X})) - \phi(g^T(\boldsymbol{M}))]\|) + \sup_{\omega \in \Omega}(\|\omega\| \, \|\mathbb{E}_{P_{00}^s}[\phi(g^T(\boldsymbol{M})) - \phi(f^T(\boldsymbol{X}))]\|) \\
&\leq \|\mathbb{E}_{P_{11}^s}[\phi(f^T(\boldsymbol{X})) - \phi(g^T(\boldsymbol{M}))]\| + \|\mathbb{E}_{P_{01}^s}[\phi(g^T(\boldsymbol{M})) - \phi(f^T(\boldsymbol{X}))]\| \\
&\quad + \|\mathbb{E}_{P_{10}^s}[\phi(f^T(\boldsymbol{X})) - \phi(g^T(\boldsymbol{M}))]\| + \|\mathbb{E}_{P_{00}^s}[\phi(g^T(\boldsymbol{M})) - \phi(f^T(\boldsymbol{X}))]\| \\
&\leq \mathbb{E}_{P_{11}^s}[L|f^T(\boldsymbol{X}) - g^T(\boldsymbol{M})|] + \mathbb{E}_{P_{01}^s}[L|g^T(\boldsymbol{M}) - f^T(\boldsymbol{X})|] \\
&\quad + \mathbb{E}_{P_{10}^s}[L|f^T(\boldsymbol{X}) - g^T(\boldsymbol{M})|] + \mathbb{E}_{P_{00}^s}[L|g^T(\boldsymbol{M}) - f^T(\boldsymbol{X})|] \\
&\leq 4L \sup_{P_{vy}^s} \mathbb{E}_{P_{vy}^s}[|g^T(\boldsymbol{M}) - f^T(\boldsymbol{X})|]
\end{aligned}$$

Following a similar approach, we can obtain a bound for MBM. The proof follows from the fact that $\mathrm{MMD}(P_{g_0^M|11}^s, P_{g_0^M|01}^s) = \mathrm{MMD}(P_{g_0^M|10}^s, P_{g_0^M|00}^s)$, as the target model $g^M$ is invariant across all distributions.

$$\begin{aligned}
\mathrm{CMMD}(P^s, g^M \circ f^M) &= P(y=1)\mathrm{MMD}(P_{g^M \circ f^M|11}^s, P_{g^M \circ f^M|01}^s) + P^s(y=0)\mathrm{MMD}(P_{g^M \circ f^M|10}^s, P_{g^M \circ f^M|00}^s) \\
&\leq \mathrm{MMD}(P_{g^M \circ f^M|11}^s, P_{g^M \circ f^M|01}^s) + \mathrm{MMD}(P_{g^M \circ f^M|10}^s, P_{g^M \circ f^M|00}^s) \\
&= [\mathrm{MMD}(P_{g^M \circ f^M|11}^s, P_{g^M \circ f^M|01}^s) - \mathrm{MMD}(P_{g_0^M|11}^s, P_{g_0^M|01}^s)] \\
&\quad + [\mathrm{MMD}(P_{g^M \circ f^M|10}^s, P_{g^M \circ f^M|00}^s) - \mathrm{MMD}(P_{g_0^M|10}^s, P_{g_0^M|00}^s)]
\end{aligned}$$

(The equality holds as $g^M$ is only a function of $\boldsymbol{M}$, which is invariant across all distributions. We can now obtain the final bounds by the L-Lipchitz and K-Lipchitz properties of the feature mapping and the target

model.)

$$\leq \sup_{\omega\in\Omega}(\mathbb{E}_{P_{11}^s}[\omega(g^M(f^M(\boldsymbol{X})))] - \mathbb{E}_{P_{01}^s}[\omega(g^M(f^M(\boldsymbol{X})))] - \mathbb{E}_{P_{11}^s}[\omega(g^M(\boldsymbol{M}))] + \mathbb{E}_{P_{01}^s}[\omega(g^M(\boldsymbol{M}))])$$

$$+ \sup_{\omega\in\Omega}(\mathbb{E}_{P_{10}^s}[\omega(g^M(f^M(\boldsymbol{X})))] - \mathbb{E}_{P_{00}^s}[\omega(g^M(f^M(\boldsymbol{X})))] - \mathbb{E}_{P_{10}^s}[\omega(g^M(\boldsymbol{M}))] + \mathbb{E}_{P_{00}^s}[\omega(g^M(\boldsymbol{M}))])$$

$$\leq \sup_{\omega\in\Omega}(\mathbb{E}_{P_{11}^s}[\omega(g^M(f^M(\boldsymbol{X})))] - \mathbb{E}_{P_{11}^s}[\omega(g^M(\boldsymbol{M}))]) + \sup_{\omega\in\Omega}(\mathbb{E}_{P_{01}^s}[\omega(g^M(\boldsymbol{M}))] - \mathbb{E}_{P_{01}^s}[\omega(g^M(f^M(\boldsymbol{X})))])$$

$$+ \sup_{\omega\in\Omega}(\mathbb{E}_{P_{10}^s}[\omega(g^M(f^M(\boldsymbol{X})))] - \mathbb{E}_{P_{10}^s}[\omega(g^M(\boldsymbol{M}))]) + \sup_{\omega\in\Omega}(\mathbb{E}_{P_{00}^s}[\omega(g^M(\boldsymbol{M}))] - \mathbb{E}_{P_{00}^s}[\omega(g^M(f^M(\boldsymbol{X})))])$$

$$\leq \sup_{\omega\in\Omega}(\langle\omega, \mathbb{E}_{P_{11}^s}[\phi(g^M(f^M(\boldsymbol{X}))) - \phi(g^M(\boldsymbol{M}))]\rangle) + \sup_{\omega\in\Omega}(\langle\omega, \mathbb{E}_{P_{01}^s}[\phi(g^M(\boldsymbol{M})) - \phi(g^M(f^M(\boldsymbol{X})))]\rangle)$$

$$+ \sup_{\omega\in\Omega}(\langle\omega, \mathbb{E}_{P_{10}^s}[\phi(g^M(f^M(\boldsymbol{X}))) - \phi(g^M(\boldsymbol{M}))]\rangle) + \sup_{\omega\in\Omega}(\langle\omega, \mathbb{E}_{P_{00}^s}[\phi(g^M(\boldsymbol{M})) - \phi(g^M(f^M(\boldsymbol{X})))]\rangle)$$

$$\leq \sup_{\omega\in\Omega}(\|\omega\| \, \|\mathbb{E}_{P_{11}^s}[\phi(g^M(f^M(\boldsymbol{X}))) - \phi(g^M(\boldsymbol{M}))]\|) + \sup_{\omega\in\Omega}(\|\omega\| \, \|\mathbb{E}_{P_{01}^s}[\phi(g^M(\boldsymbol{M})) - \phi(g^M(f^M(\boldsymbol{X})))]\|)$$

$$+ \sup_{\omega\in\Omega}(\|\omega\| \, \|\mathbb{E}_{P_{10}^s}[\phi(g^M(f^M(\boldsymbol{X}))) - \phi(g^M(\boldsymbol{M}))]\|) + \sup_{\omega\in\Omega}(\|\omega\| \, \|\mathbb{E}_{P_{00}^s}[\phi(g^M(\boldsymbol{M})) - \phi(g^M(f^M(\boldsymbol{X})))]\|)$$

$$\leq \|\mathbb{E}_{P_{11}^s}[\phi(g^M(f^M(\boldsymbol{X}))) - \phi(g^M(\boldsymbol{M}))]\| + \|\mathbb{E}_{P_{01}^s}[\phi(g^M(\boldsymbol{M})) - \phi(g^M(f^M(\boldsymbol{X})))]\|$$

$$+ \|\mathbb{E}_{P_{10}^s}[\phi(g^M(f^M(\boldsymbol{X}))) - \phi(g^M(\boldsymbol{M}))]\| + \|\mathbb{E}_{P_{00}^s}[\phi(g^M(\boldsymbol{M})) - \phi(g^M(f^M(\boldsymbol{X})))]\|$$

$$\leq \mathbb{E}_{P_{11}^s}[L \|g^M(f^M(\boldsymbol{X})) - g^M(\boldsymbol{M})\|] + \mathbb{E}_{P_{01}^s}[L \|g^M(\boldsymbol{M}) - g^M(f^M(\boldsymbol{X}))\|]$$

$$+ \mathbb{E}_{P_{10}^s}[L \|g^M(f^M(\boldsymbol{X})) - g^M(\boldsymbol{M})\|] + \mathbb{E}_{P_{00}^s}[L \|g^M(\boldsymbol{M}) - g^M(f^M(\boldsymbol{X}))\|]$$

$$\leq 4L \sup_{P_{vy}^s} \mathbb{E}_{P_{vy}^s}[\|g^M(f^M(\boldsymbol{X})) - g^M(\boldsymbol{M})\|]$$

$$\leq 4LK \sup_{P_{vy}^s} \mathbb{E}_{P_{vy}^s}[\|f^M(\boldsymbol{X}) - \boldsymbol{M}\|]$$

$\square$

**Proposition B.2** (Restated Proposition 2). *Let $\Omega$ be the unit ball in a reproducing kernel Hilbert space with an $L$-Lipchitz feature mapping used in the MMD. Also let the $g^M$ be $K$-Lipchitz, $P^s$ be some distribution that conforms to the causal DAG in Figure 1, and $P^t \in \mathcal{P}$. Then*

$$\text{For TIPMI:} \quad |R_{P^t}(f^T) - R_{P^s}(f^T)| \leq 4L \sup_{P_{vy}^s} \mathbb{E}_{P_{vy}^s}[|g^T(\boldsymbol{M}) - f^T(\boldsymbol{X})|]$$

$$\text{For MBM:} \quad |R_{P^t}(g^M \circ f^M) - R_{P^s}(g^M \circ f^M)| \leq 4LK \sup_{P_{vy}^s} \mathbb{E}_{P_{vy}^s}[\|f^M(\boldsymbol{X}) - \boldsymbol{M}\|]$$

*Proof.* This proof follows immediately from Lemma B.1 and Lemma B.2. $\square$

## C   Section 5.2.3 Proofs

**Lemma C.3.** *For any $P \in \mathcal{P}$,*

$$\|\boldsymbol{w}_\perp\| \leq \frac{CMMD(P, f)}{\|\Delta\|}.$$

*Proof.* The structure of this proof follows similar to that of Proposition 4 in (Makar et al., 2022). For $\omega \in \Omega$ where we let $\omega(\boldsymbol{x}) = \boldsymbol{w}^\top \boldsymbol{x}$,

$$
\begin{aligned}
\text{CMMD}(P, f) &\geq |\mathbb{E}[\omega(\boldsymbol{x})|v = 1, y = 1] - \mathbb{E}[\omega(\boldsymbol{x})|v = 0, y = 1]|P_s(Y = 1) \\
&\quad + |\mathbb{E}[\omega(\boldsymbol{x})|v = 1, y = 0] - \mathbb{E}[\omega(\boldsymbol{x})|v = 0, y = 0]|P_s(Y = 0) \\
&= |\mathbb{E}[\boldsymbol{w}^\top \boldsymbol{x}|v = 1, y = 1] - \mathbb{E}[\boldsymbol{w}^\top \boldsymbol{x}|v = 0, y = 1]P_s(Y = 1) \\
&\quad + \mathbb{E}[\boldsymbol{w}^\top \boldsymbol{x}|v = 1, y = 0] - \mathbb{E}[\boldsymbol{w}^\top \boldsymbol{x}|v = 0, y = 0]P_s(Y = 0)| \\
&= |\boldsymbol{w}^\top (\mathbb{E}[\boldsymbol{x}|v = 1, y = 1] - \mathbb{E}[\boldsymbol{x}|v = 0, y = 1]P_s(Y = 1) \\
&\quad + \mathbb{E}[\boldsymbol{x}|v = 1, y = 0] - \mathbb{E}[\boldsymbol{x}|v = 0, y = 0]P_s(Y = 0))| \\
&= |\boldsymbol{w}^\top \Delta|.
\end{aligned}
$$

Since $\|\boldsymbol{w}_\perp\| = \frac{|\boldsymbol{w}^\top \Delta|}{\|\Delta\|}$, we obtain the final result.

$\square$

**Definition C.1.** *Let $\boldsymbol{\epsilon} = \{\epsilon\}_{i=1}^n$ denote a vector of independent random variables where $P(\epsilon_i = 1) = P(\epsilon_i = -1) = \frac{1}{2}$. Then for the dataset $\mathcal{D} \sim P$ and the function class $\mathcal{F}$, the Rademacher complexity for a sample of size $n$ is defined as: $\mathcal{R}(\mathcal{F}) = \mathbb{E}_{\mathcal{D}} \left[\mathbb{E}_{\boldsymbol{\epsilon}} \left[\sup_{f \in \mathcal{F}} \frac{1}{n} \sum_{i=1}^n \epsilon_i f(\boldsymbol{x}_i)\right]\right]$.*

**Proposition C.3** (Restated Proposition 3). *Let $\boldsymbol{m}_\perp := \Pi \boldsymbol{x}$ and $\boldsymbol{m}_\| := (I - \Pi)\boldsymbol{x}$, and $\mathcal{R}(\mathcal{F})$ be the Rademacher complexity of some function space $\mathcal{F}$. For training data $\mathcal{D} = \{(\boldsymbol{x}_i, \boldsymbol{m}_i, y_i)\}_{i=1}^n$ where $\mathcal{D} \sim P^s$, also we have that $\sup_{\boldsymbol{m}_\perp} \|\boldsymbol{m}_\perp\|_2 \leq B_\perp$, and $\sup_{\boldsymbol{m}_\|} \|\boldsymbol{m}_\|\|_2 \leq B_\|$. Finally, let $4L \sup_{P_{vy}^s} \mathbb{E}_{P_{vy}^s}[|g^T(\boldsymbol{M}) - f^T(\boldsymbol{X})|] \leq \tau$ and $4LK \sup_{P_{vy}^s} \mathbb{E}_{P_{vy}^s}[|f^M(\boldsymbol{X}) - \boldsymbol{M}|] \leq \tau'$. Then*

$$
\mathcal{R}(\mathcal{F}_{L2}) \leq \frac{A\sqrt{B_\|^2 + B_\perp^2}}{\sqrt{n}} \quad, \quad \mathcal{R}(\mathcal{F}_{TIPMI}) \leq \frac{A \cdot B_\| + \tau \frac{B_\perp}{\|\Delta\|}}{\sqrt{n}} \quad and \quad \mathcal{R}(\mathcal{F}_{MBM}) \leq \frac{A \cdot B_\| + \tau' \frac{B_\perp}{\|\Delta\|}}{\sqrt{n}}
$$

*Proof.* First, we derive the bound on $\mathcal{R}(\mathcal{F}_{L2})$, which follows directly from Proposition 5 from Makar et al. (2022):

$$
\begin{aligned}
\mathcal{R}(\mathcal{F}_{L2}) &= \mathbb{E}_{\mathcal{D}} \mathbb{E}_{\boldsymbol{\epsilon}} [\sup_{\boldsymbol{w}:\|\boldsymbol{w}\|_2 \leq A} \frac{1}{n} \sum_i \epsilon_i \boldsymbol{w}^T \boldsymbol{x}_i] \\
&= \mathbb{E}_{\mathcal{D}} \mathbb{E}_{\boldsymbol{\epsilon}} [\sup_{\boldsymbol{w}:\|\boldsymbol{w}\|_2 \leq A} \frac{1}{n} \sum_i \epsilon_i \boldsymbol{w}^T ((I - \Pi)\boldsymbol{x}_i + \Pi \boldsymbol{x}_i)] \\
&= \mathbb{E}_{\mathcal{D}} \mathbb{E}_{\boldsymbol{\epsilon}} [\sup_{\boldsymbol{w}:\|\boldsymbol{w}\|_2 \leq A} \frac{1}{n} \sum_i \epsilon_i \boldsymbol{w}^T (\boldsymbol{m}_{\|i} + \boldsymbol{m}_{\perp i})] \\
&\leq \frac{A\sqrt{B_\|^2 + B_\perp^2}}{\sqrt{n}}
\end{aligned}
$$

Next, we derive the bounds for $\mathcal{R}(\mathcal{F}_{TIPMI})$ and $\mathcal{R}(\mathcal{F}_{MBM})$. For some function class $\mathcal{F}$, we get that

$$\mathcal{R}(\mathcal{F}) = \mathbb{E}_{\mathcal{D}}\mathbb{E}_{\epsilon}\big[\sup_{\boldsymbol{w}:\|\boldsymbol{w}\|_2 \leq A} \frac{1}{n}\sum_i \epsilon_i \boldsymbol{w}^T \boldsymbol{x}_i\big]$$

$$= \mathbb{E}_{\mathcal{D}}\mathbb{E}_{\epsilon}\big[\sup_{\boldsymbol{w}:\|\boldsymbol{w}\|_2 \leq A} \frac{1}{n}\sum_i \epsilon_i (\Pi\boldsymbol{w}^T \boldsymbol{x}_i + (I-\Pi)\boldsymbol{w}^T \boldsymbol{x}_i)\big]$$

$$\leq \mathbb{E}_{\mathcal{D}}\mathbb{E}_{\epsilon}\big[\sup_{\substack{\boldsymbol{w}_{\|}:\|\boldsymbol{w}_{\|}\|_2 \leq A \\ \boldsymbol{w}_{\perp}:\|\boldsymbol{w}_{\perp}\|_2 \leq A}} \frac{1}{n}\sum_i \epsilon_i (\boldsymbol{w}_{\perp}^T \boldsymbol{m}_{\perp i} + \boldsymbol{w}_{\|}^T \boldsymbol{m}_{\|i})\big]$$

$$\leq \mathbb{E}_{\mathcal{D}}\mathbb{E}_{\epsilon}\big[\sup_{\boldsymbol{w}_{\perp}:\|\boldsymbol{w}_{\perp}\|_2 \leq A} \frac{1}{n}\sum_i \epsilon_i \boldsymbol{w}_{\perp}^T \boldsymbol{m}_{\perp i}\big] + \mathbb{E}_{\mathcal{D}}\mathbb{E}_{\epsilon}\big[\sup_{\boldsymbol{w}_{\|}:\|\boldsymbol{w}_{\|}\|_2 \leq A} \frac{1}{n}\sum_i \epsilon_i \boldsymbol{w}_{\|}^T \boldsymbol{m}_{\|i}\big]$$

$$\leq \frac{A \cdot B_{\|} + \mathrm{CMMD}(P^s, f)\frac{B_{\perp}}{\|\Delta\|}}{\sqrt{n}}$$

The final bounds for both $\mathcal{R}(\mathcal{F}_{\mathrm{TIPMI}})$ and $\mathcal{R}(\mathcal{F}_{\mathrm{MBM}})$ follow directly from Lemma B.2.

$\square$

# D   Datasets

## D.1   Waterbirds

The waterbirds dataset, which was first introduced by (Sagawa et al., 2019), uses labeled images of birds and their segmentations from the CUB-200-2011 dataset (Wah et al., 2011). If the bird is an Albatross, Auklet, Cormorant, Frigatebird, Fulmar, Gull, Jaeger, Kittiwake, Pelican, Puffin, Tern, Gadwall, Grebe, Mallard, Merganser, Guillemot, or a Pacific Loon, we classify it as a waterbird; every other bird from the dataset we classify as a landbird. We use a subset of the places dataset (Zhou et al., 2017), for the background images. Specifically, we use the 200 land backgrounds and 300 water backgrounds provided by Makar et al. (2022). Similar to (Makar et al., 2022), we derive other backgrounds from these images by applying rotations, manipulating brightness, and zooming in. The manipulated images are obtained from (Makar et al., 2022). In total, our waterbirds dataset is comprised of 8,672 landbirds and 2,483 waterbirds. The bird images, along with the background images, are randomly split so that 80% of the images are used to develop the synthetic training sets, and 20% are used for the testing sets.

The TIPMI teacher and MBM mediator models are trained on the segmented images of birds with a black background, which is the privileged mediation information, as is shown in Figure 6. The student models are trained on images of birds containing either land or water backgrounds. Background and bird images are only used once to generate samples for the training and testing datasets. The size of the images is $256 \times 256$ pixels, except for the finite-sample efficiency experiments, which use $128 \times 128$. Example images taken from one of the generated waterbirds datasets are shown in Figure 7. For an additional shortcut, we add small black squares randomly placed within an image to simulate camera artifacts. An example of a landbird with this shortcut is given in Figure 8.

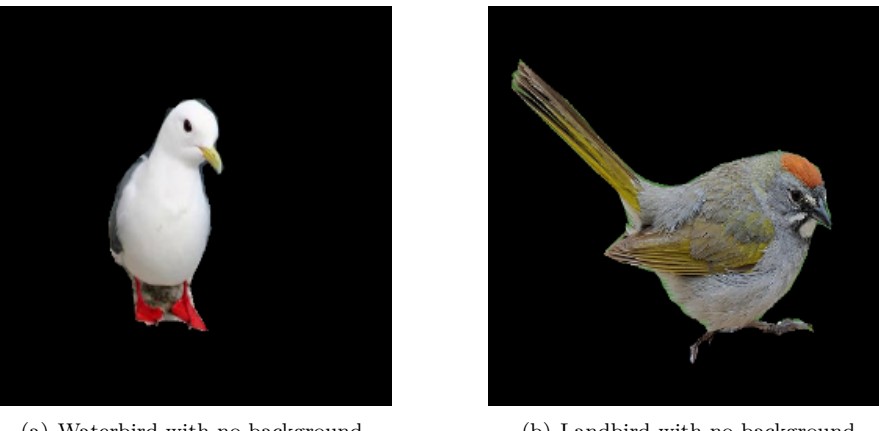

(a) Waterbird with no background    (b) Landbird with no background

Figure 6: An example of the TIPMI teacher and MBM mediator models training data for the waterbirds experiments. It consists of waterbird and landbird images with black backgrounds.

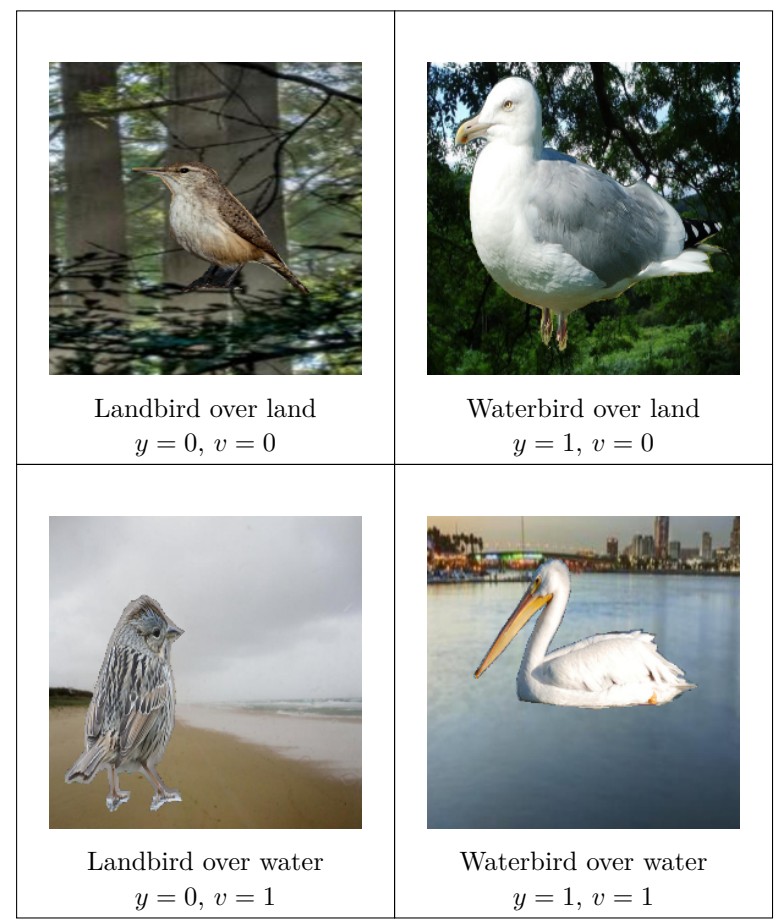

Figure 7: Examples of generated images used in the waterbirds dataset, where the only shortcut is the background.

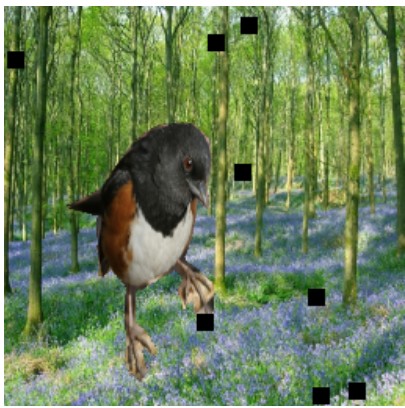

Figure 8: Example image from waterbirds dataset with two shortcuts. The first shortcut is the background, and the second shortcut is the black squares that simulate camera artifacts.

### D.2 Food Review

The food review dataset is created from 20,000 randomly selected reviews from the Amazon Food Review dataset (McAuley & Leskovec, 2013). It contains reviews, the amount of stars (1-5), and a summary of each review. Each synthetic training dataset contains 16,000 samples, whereas the testing set contains 4,000. To inject a shortcut into the reviews, we add perturbation to the words "the" and "be" such that "the" becomes "thexxxxx" and "be" becomes "bexxxxx", similar to what is done in (Veitch et al., 2021). For instance, the review "They make the best gummies hands down" is turned into "They make thexxxxx best gummies hands down". We add a similar shortcut by adding perturbations to the words "a" and "to" such that "a" becomes "ayyyyy" and "to" becomes "toyyyyy". The labels are binarized such that $Y = 0$ for 1-3 stars and $Y = 1$ for 4-5 stars. We employ a summary of the review as the mediator, generated by a Llama-2-7b-chat model (Touvron et al., 2023). These model-generated summaries replace the original ones in the Amazon Food Review dataset, as many of the latter were not sufficiently informative.

### D.3 KOA

The knee osteoarthritis dataset (Nevitt et al., 2006) is comprised of knee X-ray images and joint space width measurements from the Osteoarthritis Initiative, which is publicly available for download at https://nda.nih.gov/oai. The knee X-ray images are obtained from the "OAI12MonthImages" sub-dataset. They are stored as a DICOM image and include both of the patient's knees. We convert each image to a PNG and split it in half so that each image only contains a patient's left or right knee. Each image is then centered and resized to 256x256 pixels. The training dataset is comprised of 80% of the total 4982 samples, whereas the testing set is the other 20%. The datasets are split so that one patient's X-rays could only be in one of the two datasets. To determine if a knee in the X-ray had osteoarthritis, we use the Kellgren-Lawrence (KL) grades (0-4) (Kohn et al., 2016) that were provided for each knee. If the knee had a grade of 2, 3, or 4, we classified the sample as having OA (Y=1); if it had a grade of 0 or 1, we classified it as normal (Y=0). For final evaluation, we discard all samples with a KL grade of 2 as it is vaguely defined, and KOA diagnoses labeled with this grade are subjective (Olsson et al., 2021). The distribution of the KL grades is provided in Table 3.

To inject spurious correlations into the dataset, we overlay a black box over the X-rays such that the presence of the black box was correlated with an OA diagnosis. This was done to mimic a metallic token, which may act as a spurious feature in real-world X-ray classification problems (Zech et al., 2018). Examples of the spuriously correlated KOA dataset are shown in Figure 9. To evaluate how well TIPMI, MBM, and the baselines perform when multiple spurious features are present, we also create a dataset with an additional spuriously correlated white box, as is shown in Figure 10. For the privileged mediation information, we use 16 joint space width measurements that are provided with each knee X-ray.

Table 3: The distribution of KL grades in the KOA dataset.

| KL Grade | Number of Samples |
|---|---|
| 0 | 978 |
| 1 | 671 |
| 2 | 1928 |
| 3 | 1082 |
| 4 | 323 |

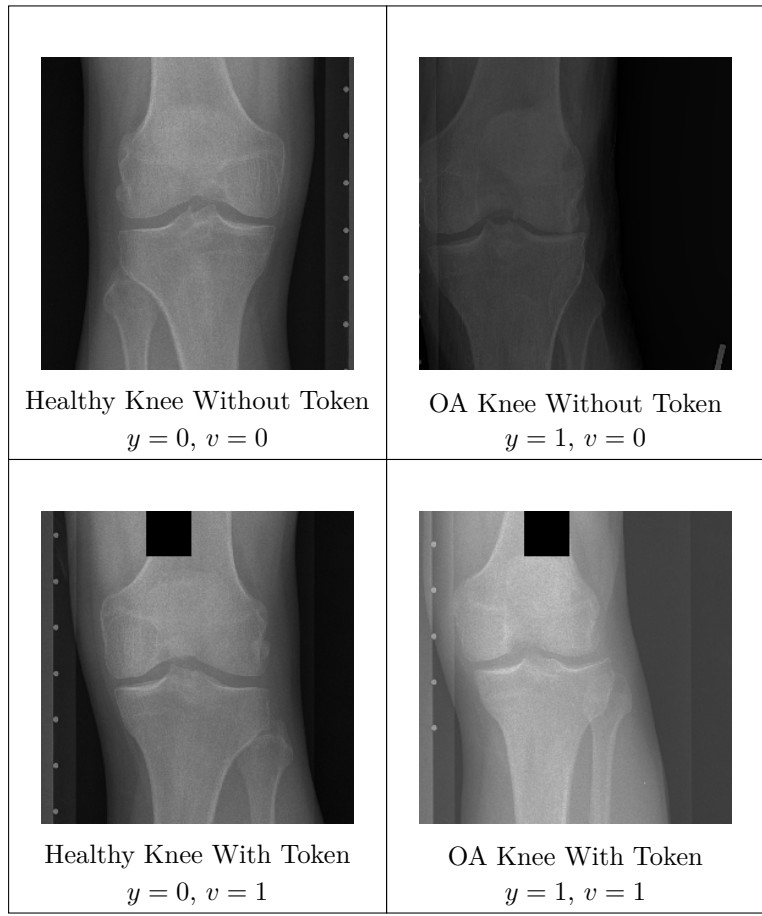

Figure 9: Examples of generated images used in the KOA dataset with a single shortcut.

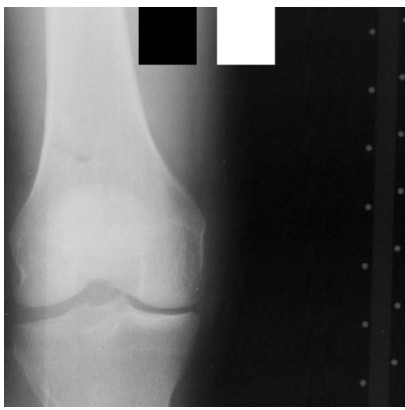

Figure 10: Example image from KOA dataset with two shortcuts represented by a white and black square spuriously associated with osteoarthritis.

# E  Experiment Setup

**Models.**  For the waterbirds experiments, the TIPMI teacher models, TIPMI student models, MBM target models, and all baseline models are ResNet-50 models (He et al., 2016) pre-trained on ImageNet-1k (Russakovsky et al., 2015). The MBM mediator model is a U-Net with a ResNet-50 encoder, which is also pre-trained on ImageNet-1k For KOA, the TIPMI student, MBM mediator models, and all baseline models are ResNet-50 models, and the TIPMI teacher and MBM target models are single-layer neural networks with 1024 hidden units. For the food review experiments, the TIPMI teacher model, student model, MBM target model, and baseline models are BERT-tiny classifiers (Bhargava et al., 2021), and the MBM mediator model is a T5-small model (Raffel et al., 2020). sAll models use the cross entropy loss, except for the TIPMI teacher models, which use the mean squared error, and the MBM mediator model used in the waterbirds dataset, which uses the dice loss.

**TIPMI Implemenation.**  The TIPMI teacher was implemented as described in Section 4, "Teacher (Step 1)". However, the student was trained by using the MSE to match the student logits with the teacher logits, not the final probabilities.

**MBM Implementation.**  We train MBMs sequentially, similar to sequential bottleneck models as described in (Koh et al., 2020): we first train the mediator model $f^M(\boldsymbol{X})$ and then train the target model $g^M(\boldsymbol{M})$ using the mediator predictions of $f^M(\boldsymbol{X})$.

**Hyperparameter Selection.**  The hyperparameters for all models are chosen using cross-validation across the 10 simulations. For GDRO, SCPM, and IRM, the hyperparameters are chosen based on the highest worst-group accuracy. The hyperparameters for the MBM mediator model were chosen based on the lowest MSE for KOA, highest ROUGE-1 score in food review, and lowest DICE loss in waterbirds. For all other models, the hyperparameters were chosen based on which model had the best AUROC.

For the waterbirds experiments, we perform cross-validation with a learning rate of $1e^{-5}$ and $L_2$ penalty parameters with the values $[0, 1e^{-3}, 1e^{-5}]$ for all models except SCPM, MCPM, and IRM. For SCPM, we set the learning rate to $1e^{-5}$, and cross-validate over $\sigma = [1e1, 1e2]$ and $\alpha = [1e0, 1e1, 1e2]$. For MCPM, we set the learning rate to $1e^{-5}$ and cross-validate over $\alpha = [1e0, 1e1, 1e2]$. $\sigma$ is chosen via the median heuristic (Garreau et al., 2017). For IRM, we set the learning rate to $1e^{-5}$ and cross-validate over $\alpha = [1e-1, 1e0, 1e1]$. We train each model using Adam (Kingma & Ba, 2014) and a batch size of 32, except for SCPM and MCPM, which use a batch size of 256. Each model is trained for 100 epochs.

For the food review experiments, we perform cross-validation with a learning rate of $1e^{-5}$ and $L_2$ penalty parameters with the values $[0, 1e^{-3}, 1e^{-5}]$ for all models except SCPM, MCPM, and IRM. For SCPM, we set the learning rate to $1e^{-5}$, and cross-validate over $\sigma = [1e1, 1e2]$ and $\alpha = [1e0, 1e1, 1e2, 1e3]$. For MCPM, we

set the learning rate to $1e^{-5}$ and cross-validate over $\alpha = [1e0, 1e1, 1e2]$. $\sigma$ is chosen via the median heuristic (Garreau et al., 2017). For IRM, we set the learning rate to $1e^{-5}$ and cross-validate over $\alpha = [1e-1, 1e0, 1e1]$. We train each model using Adam and a batch size of 32, except for SCPM and MCPM, which use a batch size of 256. Each model is trained for 10 epochs, except for SCMP and MCPM, which are trained for 20 epochs.

For the KOA experiments, we perform cross-validation with a learning rate of $1e^{-4}$ and $L_2$ penalty parameters with the values $[0, 1e^{-3}, 1e^{-5}]$ for all models except SCPM, MCPM, and IRM. For SCPM, we set the learning rate to $1e^{-4}$, and cross-validate over $\sigma = [1e1, 1e2]$ and $\alpha = [1e0, 1e1, 1e2, 1e3]$. For MCPM, we set the learning rate to $1e^{-5}$ and cross-validate over $\alpha = [1e0, 1e1, 1e2]$. $\sigma$ is chosen via the median heuristic (Garreau et al., 2017). For IRM, we set the learning rate to $1e^{-4}$ and cross-validate over $\alpha = [1e-1, 1e0, 1e1]$. We train each model using Adam and a batch size of 32, except for SCPM and MCPM, which use a batch size of 256. Each model is trained for 100 epochs.

**Training Environment.** All models used throughout this paper are implemented in PyTorch (Paszke et al., 2019). We train each model using a Nvidia A40 GPU and 36 GB of memory on a Linux operating system. All experiments took approximately a month with three Nvidia A40 GPUs.

# F  Additional Empirical Results

## F.1  Two shortcuts

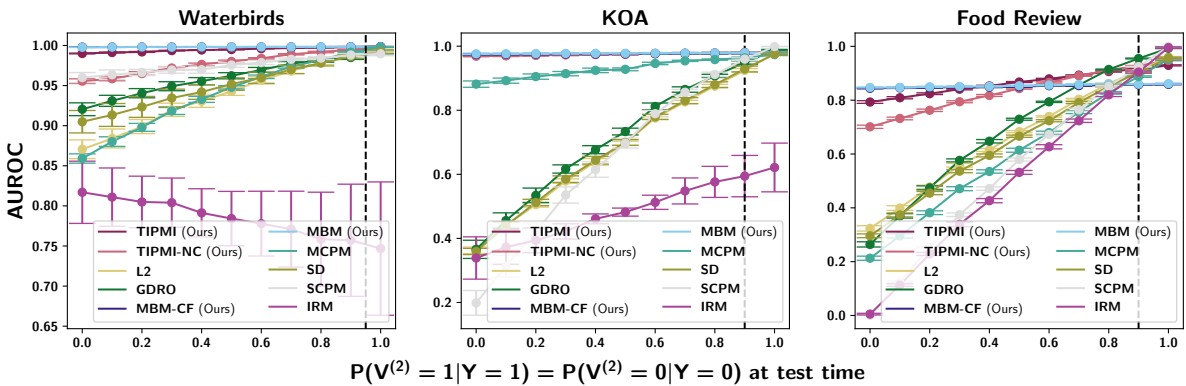

Figure 11: $x$-axis shows $P(V^{(2)}|Y)$ at test time. $y$-axis shows the AUROC over the test data. Spurious correlations are introduced during training by setting $P_s(V^{(d)} = 1|Y = 1) = P_s(V^{(d)} = 0|Y = 0) = 0.95$ in the waterbirds dataset and $P_s(V^{(d)} = 1|Y = 1) = P_s(V^{(d)} = 0|Y = 0) = 0.9$ in the food review and KOA datasets, for each shortcut $V^{(1)}$ and $V^{(2)}$.

## F.2  Single shortcut with overlap

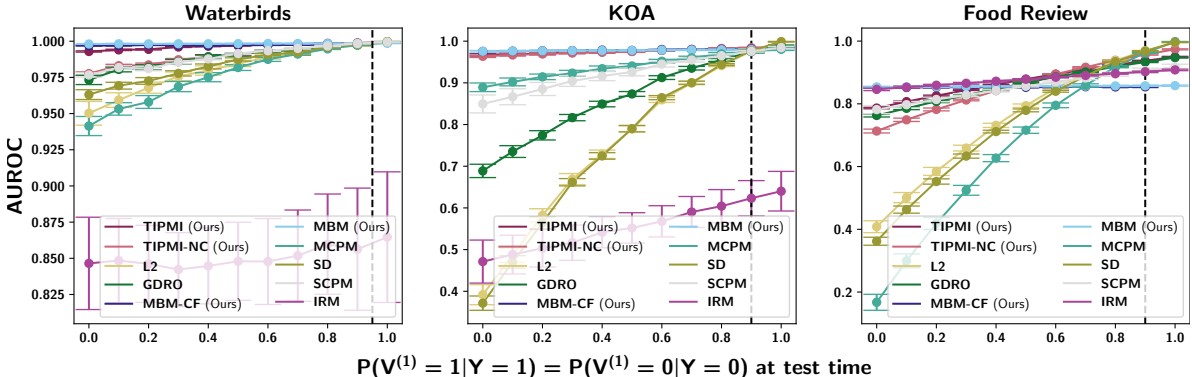

Figure 12: The x-axis shows $P(Y|V^{(1)})$ at test time under different shifted distributions, the y-axis shows the AUROC over the test data, and the dashed vertical line shows $P(Y|V^{(1)})$ at training time. Spurious correlations are introduced during training by setting $P_s(V^{(1)} = 1|Y = 1) = P_s(V^{(1)} = 0|Y = 0) = 0.95$ in the waterbirds dataset and $P_s(V^{(1)} = 1|Y = 1) = P_s(V^{(1)} = 0|Y = 0) = 0.9$ in the food review and KOA datasets at training.

### F.3 Single shortcut with no overlap

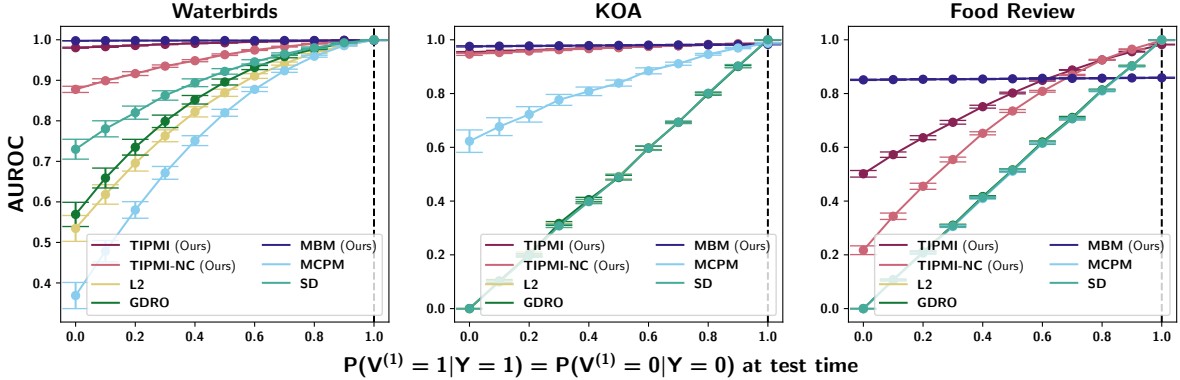

Figure 13: The x-axis shows $P(Y|V^{(1)})$ at test time under different shifted distributions, the y-axis shows the AUROC over the test data, and the dashed vertical line shows $P(Y|V^{(1)})$ at training time. Spurious correlations are introduced during training by setting $P_s(V^{(1)} = 1|Y = 1) = P_s(V^{(1)} = 0|Y = 0) = 1.0$ for all datasets at training.

### F.4 Finite-sample efficiency results under the unconfounded distribution

To isolate the gains in finite-sample efficiency introduced by TIPMI and MBM, we run an "in-distribution" experiment where the training and test data are drawn from the unconfounded distribution, $P^{\circ}$, where $V^{(1)}$ is simulated such that $V^{(1)} \perp\!\!\!\perp Y$. The results are shown in Table 4. TIPMI and MBM outperform all baselines over the waterbirds and KOA datasets. However, for food review, TIPMI-NC, L2, and GRDO perform the best. The difference between TIPMI-NC and TIPMI is likely due to the TIPMI teacher models having less training data from the cross-fitting procedure.

### F.5 Single shortcut setting where the shortcut affects the mediator

Table 4: In distribution results.

| Model \ Data | Waterbirds | KOA | Food Review |
|---|---|---|---|
| TIPMI | **0.992 (0.002)** | **0.980 (0.004)** | **0.896 (0.007)** |
| TIPMI-NC | 0.983 (0.005) | **0.980 (0.007)** | **0.908 (0.005)** |
| L2 | **0.987 (0.004)** | 0.946 (0.010) | **0.907 (0.006)** |
| GDRO | **0.986 (0.006)** | 0.948 (0.008) | **0.903 (0.007)** |
| MCPM | 0.985 (0.004) | 0.948 (0.008) | **0.899 (0.007)** |
| MBM | **0.993 (0.003)** | **0.980 (0.006)** | 0.857 (0.005) |
| MBM-CF | **0.993 (0.003)** | **0.977 (0.007)** | 0.853 (0.006) |

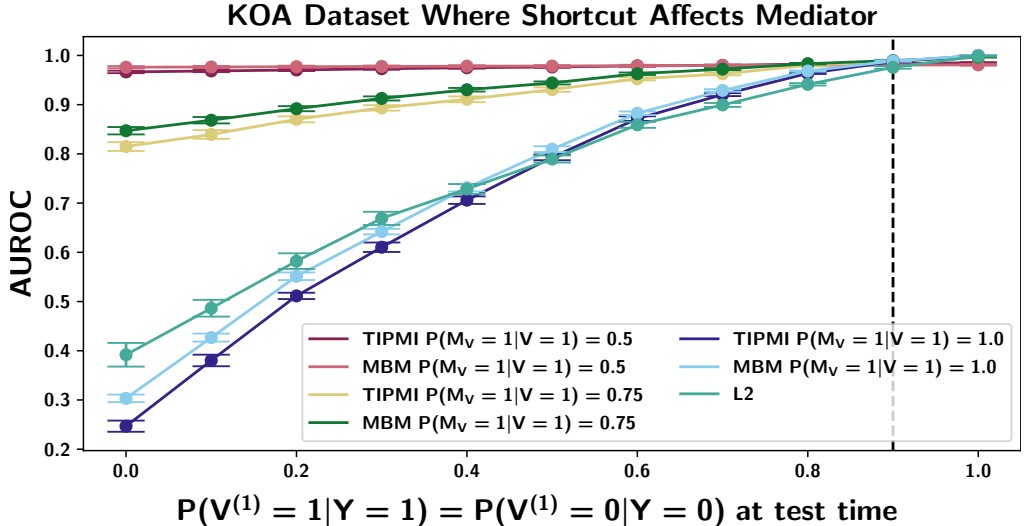

Figure 14: The x-axis shows $P(Y|V^{(1)})$ at test time under different shifted distributions, the y-axis shows the AUROC over the test data, and the dashed vertical line shows $P(Y|V^{(1)})$ at training time. Spurious correlations are introduced during training by setting $P_s(V^{(1)} = 1|Y = 1) = P_s(V^{(1)} = 0|Y = 0) = 0.9$ for all datasets at training.

In this section, we examine how sensitive TIPMI and MBM to varying degrees of causal influence from the shortcut to the mediator. For these experiments, we added an additional binary variable $M_V$ to the mediator in the KOA dataset that may be causally influenced by a single shortcut. Spurious correlations are introduced during training by setting $P_s(V^{(1)} = 1|Y = 1) = P_s(V^{(1)} = 0|Y = 0) = 0.9$ in the KOA dataset and at test time we vary $P_t(V|Y)$ to assess sensitivity to the unobserved shortcut.

We examined MBM and TIPMI in several settings where we varied the amount of influence of $V$ onto $M_V$: one where $P_s(V_1 = 1|M_V = 1) = P_s(V_1 = 0|M_V = 0) = 0.5$, $P_s(V_1 = 1|M_V = 1) = P_s(V_1 = 0|M_V = 0) = 0.75$, and $P_s(V_1 = 1|M_V = 1) = P_s(V_1 = 0|M_V = 0) = 1.0$. Our results are presented in Figure F.5, where we showed that as long as the influence from $V$ to $M_V$ is not strong, both TIPMI and MBM will still outperform standard L2 regularization.

