# OpenReview forum: "Teaching Invariance Using Privileged Mediation Information"
_TMLR — Accepted by TMLR_

### Review · Reviewer_rRJz · 2025-10-17

**Summary Of Contributions:**

This paper describes two novel approaches for leveraging mediators to achieve better out-of-distribution generalization, compared to relying on known learning shortcuts. It relies on a set of annotated mediator variables that are assumed independent of potential shortcuts and distribution shifts. The first proposed approach uses one model to learn the mediators from the input features, and another model to learn the targets from these predicted mediators. This approach provides the strongest robustness and generalization results. However, the two models lead to increased inference resource demands. To address this, the second proposed approach first learns an auxiliary teacher model that predicts targets from ground-truth mediators. A second model then imitates this teacher model, mapping from inputs to the target predicted by the teacher. Only the second model is required for inference.

**Additional Comments:**

**Typos:**
- On page 3, the introduction "PI" is used, that was not introduced before
- Since Figure 1 is a name, like Lemma 1, it should be capitalized. I also saw this for Proposition 2.
- Page 7, bottom: "The orthogonal projection *is* as X *is* only". One *is* too much?
- Abbreviations are not capitalized correctly in the References section. For example: "... augmentations for text *ood* generalization ...".

**Audience:**

Yes

**Audience Explanation:**

The contributions in this paper are relevant to a broad audience, since distribution shifts are ample in real-world applications.

**Broader Impact Concerns:**

I have no broader impact concerns.

**Claims And Evidence:**

Yes

**Claims Explanation:**

**Caveat: I was unable to check the proofs, since this paper is outside my research area. For the same reason, I can not judge whether the selected baselines are sufficient.**

For the most part, the paper supports its claims well, as far as I can judge. The paper includes a strong experimental evaluation, comparing to several baselines on three datasets with different data modalities.

I have an issue with the restriction to two shortcut confounders in Section 3.1. While Section 3.1 claims that the "results hold for [confounders] V of arbitrary dimension",  the theoretical results in the paper, including the proofs in the appendix, are for two shortcut confounders. For the presentation, it is reasonable to assume two confounders; however, the theoretical results should apply to an arbitrary number to support the above claim.

**Requested Changes:**

**Required:**
1. Please clarify the issue with the theoretical contribution I outlined above.
2. Section 3.1 introduces the assumption that the mediators M are determined entirely by the input features X. This is a strong assumption that should be discussed in more detail. It should also be mentioned in the introduction.
3. I do not understand your discussion on overfitting in Section 4.3. Please elaborate: the setting is that the teacher model overfits the training data, while observing only the mediators. Clearly, that can cause the student also to overfit, but you describe this as diminishing the benefits of distillation with regard to avoiding shortcuts. As far as I can see, overfitting can only cause the student to rely on shortcuts if the mediators are confounded with the shortcuts. However, this would violate your assumption that the shortcuts do not influence the mediators.
4. Please elaborate: In Section 6.3, you state that TIPMI and MBM perform better than the baselines because they do not rely on expensive penalty terms. Why does the cost of penalty terms influence their performance in terms of AUROC?
5. In Section 6.4, you compare the model sizes of TIPMI and MBM. Your comparison does not show that larger models are required for MBM. Please also report the performance of MBM using models of comparable size to TIPMI in this table.
6. I do not understand why it is an issue for performance to predict superfluous mediators in Figure 5.
7. Please refer to and link to your proofs in the Appendix in the main body of the paper.

**Strengthens the Paper:**
The paper is already well-written. For a few paragraphs, I have suggestions for further improving the presentation:
- In the last paragraph of the introduction, you write that TIPMI is advantageous, since it does not require modelling the potentially complex mapping from X to M. However, TIPMI requires modelling the mapping from X to Y instead. Instead, the core advantage of TIPMI is lower inference cost.
- I did not understand from the notation why Equation (1) is a set of multiple elements.
- In Teacher (Step 1), item 1, it would help to state that $I_1, \ldots, I_k$ are sets of indices, for example $I_1, \ldots, I_k \subset \{1, \ldots, N\}$.
- Instead of using (1) and (2) when enumerating in text, such as in Proposition 1, I recommend using (a), (b) or (i), (ii). I was confused whether (1) and (2) referred to the equations.
- Otherwise, the forward reference to the ablation study in the last paragraph on page 9 is clunky. Perhaps place the ablation study before the robustness study.
- Figure titles in Section 6 would help to understand the figures.

---

> ### Author Response · Authors · 2025-11-07
> **Assumptions, TIPMI overfitting, and experiments**
>
> We thank the reviewer for their thoughtful and constructive feedback!
>
> **1 - Theory only considers a single binary shortcut.**
>
> We chose the setting of a single unknown binary shortcut in Section 3.1 for clarity and simplicity. However, the theoretical arguments can extend to the case of multiple shortcut variables. Our theory follows from Makar et al. [1], and the same extension to multiple confounders used in Makar et al. [2] applies here. We also emphasize that our empirical results already operate in settings with multiple unknown shortcuts and demonstrate that both MBM and TIPMI remain robust when multiple shortcuts are present.
>
> **2 - Mediators can be recovered entirely by the input.**
>
> We have revised Section 3.1 and the introduction to provide a more detailed discussion of the assumption that mediators are fully recoverable from the input feature $\boldsymbol{X}$. This assumption follows prior work in robustness and causally motivated methods (e.g., Makar et al. [1]), where mediators are modeled as deterministic functions of the inputs to enable clear theoretical analysis for the proof of Proposition 1. We also note that this assumption is typically satisfied in settings where the input features are rich, unstructured data, such as image or text data.
>
> **3 - TIPMI overfitting**
>
> The main issue with overfitting arises when the teacher model memorizes the training data and produces predictions that are nearly identical to the ground-truth labels $Y$. In this case, the student effectively learns to map $X \rightarrow Y$ directly, rather than inheriting the invariance properties of the teacher, making it similarly susceptible to shortcuts as a standard training setup that maps $X \rightarrow Y$. We have clarified this further in the revised version of the paper.
>
> **4 - TIPMI and MBM do not rely on expensive penalty terms**
>
> In our theoretical analysis, we show that TIPMI and MBM achieve improved finite-sample efficiency. While baseline methods such as SCPM also aim to enforce invariance through regularization, they rely on nonparametric conditional independence penalties that require large sample sizes for accurate estimates, which reduces their practical efficiency with finite data. We have clarified this in the revised manuscript.
>
> **5 - Comparable model sizes of TIPMI and MBM**
>
> Our goal in this section is to emphasize that MBM inherently requires larger models than TIPMI, because MBM must deploy both a target model and a mediator model at inference time, whereas TIPMI only deploys a student model. Thus, when the target and student architectures are the same (e.g., ResNet-50 for the Waterbirds and BERT-Tiny for the Food Review experiments), MBM will necessarily require more parameters at inference due to the additional mediator model. We clarify this in the updated version of the paper.
>
> For the Food Review experiments, MBM requires a pretrained text-to-text mediator model (T5-small in our case). The smallest publicly available model of this class we identified is T5-Tiny (16M parameters), which is still roughly 4x larger than the TIPMI student model used in our experiments. Pretraining a smaller text-to-text model from scratch is beyond our computational resources, so constructing a mediator model comparable in size to the TIPMI student is not feasible in practice.
>
> **6 - Performance decrease for MBM in Figure 5**
>
> The performance drop arises because the mediator model in MBM introduces additional estimation noise when predicting $M$, especially as the number of redundant mediator dimensions increases. This compounded noise reduces the quality of the signal available for the target model to predict the label. We now discuss this in more detail in the revised manuscript.
>
> **7 - Link to proofs in the Appendix**
>
> We have updated the paper to include proper links to the corresponding proofs in the Appendix.
>
> **8 - Suggestions to strengthen the paper**
>
> We thank the reviewer for their helpful suggestions to improve the clarity of our paper. We have incorporated recommendations in the revised manuscript.
>
> **9 - Typos**
>
> We thank the reviewer for catching these typos. We have updated the paper to fix these typos.
>
> **Citations**
>
> [1] Makar, Maggie, et al. "Causally motivated shortcut removal using auxiliary labels." International Conference on Artificial Intelligence and Statistics. PMLR, 2022.
>
> [2] Zheng, Jiayun, and Maggie Makar. "Causally motivated multi-shortcut identification and removal." Advances in Neural Information Processing Systems 35 (2022)

---

> > ### Comment · Reviewer_rRJz · 2025-11-08
> > **Answer to Authors**
> >
> > Thank you for addressing several of the points I brought up. Thank you for your clarifications on my requests 2, 4, 6 and 7. I agree with the points you made for these requests. I have further questions regarding the remaining requests I made.
> >
> > 1. I think it is a good idea to work with two confounders for the main part of the paper. However, this is not the case for the theoretical results. Since you "stress that [your] results hold for V of arbitrary dimension" in Section 3.1, I would like to see that proven in the paper. If there are already extensions to the analysis you base your results on for more confounders, then that should be all the more reason to state your results for more than two confounders. The same holds for your remark on the experimental setting. If your current results that are stated for two confounders do not apply in your experimental setting, this is all the more reason to prove your results in the more general setting.
> >
> > 5. My point is that MBM does not inherently require a larger model at inference time than TIPMI. As a simple example, if TIPMI uses a 1M parameter model, one could use a 500K mediator and a 500K target model for MBM and obtain the same inference speed for both methods. Of course, that might impact accuracy, but this is not clear a priori. You results on KOA show that MBM can match TIPMI in accuracy at a similar inference cost.
> >
> > 3. Thank you for the explanation how overfitting can arise. I did not yet understand the relation between overfitting and shortcuts. As I wrote: "As far as I can see, overfitting can only cause the student to rely on shortcuts if the mediators are confounded with the shortcuts." I would be grateful if you could elaborate on this, although this is a more minor point.

---

> > > ### Comment · Reviewer_rRJz · 2025-11-27
> > > **Temporary Recommendation**
> > >
> > > I have been asked by OpenReview to submit an official recommendation on this paper. Right now, a couple of points are still open, so I can not give a final recommendation.
> > >
> > > That being said, if the AE wished to accelerate the process, my recommendation right now would be "learning accept" since most of my points have been addressed. The reason I can not fully endorse this paper is point 1 in the "Answer to Authors" above.

---

> > > > ### Comment · Action_Editor_H2bn · 2025-12-19
> > > >
> > > > Hi, I would like to close the process. Has the first point been addressed? Either way, if you could submit the leaning acceptance recommendation, I would have enough information to make the decision.
> > > >
> > > > Thanks!

---

### Review · Reviewer_dzPR · 2025-10-22

**Summary Of Contributions:**

This paper addresses the critical problem of shortcut learning, where models rely on spurious correlations in training data and lead to poor out-of-distribution performance. Instead of requiring privileged information about the shortcuts themselves, which may be unknown or hard to label, the authors proposed a novel perspective to leverage the privileged mediation information, which are intermediate variables that lie on the true causal path from the input to the label and are assumed to be available only during training.

Based on this idea, the paper introduces two methods: (1) Mediator Bottleneck Models (MBMs): An adaptation of Concept Bottleneck Models, which first trains a model to predict the mediator from the input features and then a second model to predict the final label from this predicted mediator. (2) Teaching Invariance using Privileged Mediation Information (TIPMI): A knowledge distillation framework where an invariant teacher is first trained to predict the label using only the privileged mediator, and a student model is then trained on the standard inputs to fit the teacher's predictions, thereby inheriting its invariance properties.

This paper further provides a theoretical analysis that establishes the proposed methods' ability to achieve counterfactual invariance. This paper also analyzes their generalization properties, showing they can achieve tighter Rademacher complexity bounds compared to standard regularization or penalty-based methods. Numerical experiments are also provided to support the proposed methods.

**Audience:**

Yes

**Audience Explanation:**

The work focuses on the robustness under distribution shift, causality, and learning with privileged information, which are core interests for TMLR’s audience.

**Broader Impact Concerns:**

None.

**Claims And Evidence:**

Yes

**Claims Explanation:**

The paper’s claims are clearly stated and supported by theoretical analysis. Numerical results comparing the proposed methods and baseline models on multiple datasets and the ablation study on the benefits of cross-fitting for both TIPMI and MBM are also consistent with the claims.

**Requested Changes:**

(1) This paper assumes that the shortcuts do not affect the mediator, which can be a strong assumption in practice. To show the robustness of the proposed algorithms, it is meaningful to present the empirical performance of MBM and TIPMI when there is a weak correlation between the shortcuts and the mediator.

(2) The paper considers the scenario where there is a single binary shortcut. The authors can further explain how the analysis can be extended to multiple shortcuts and show the dependency of the error bounds on the number of shortcuts.

---

> ### Author Response · Authors · 2025-11-07
> **Assumption that shortcuts do not affect the mediator and theory**
>
> We thank the reviewer for their thoughtful and constructive feedback!
>
> **1 -- Assumption that shortcuts do not affect the mediator.**
>
> We agree that, in practice, mediators may be partially influenced by shortcuts. To address this, we have added new experiments (see Appendix F) that examine the sensitivity of MBM and TIPMI to varying degrees of influence from the shortcut to the mediator, using the KOA dataset. Our results show that both methods remain robust and continue to outperform baseline approaches as long as the shortcut's effect on the mediator is moderate.
>
> **2 -- Theory only considers a single binary shortcut.**
>
> We chose the setting of a single unobserved binary shortcut in Section 3.1 for clarity and simplicity. However, the theoretical arguments can extend to the case of multiple shortcut variables. Our theory follows from Makar et al. [1], and the same extension to multiple confounders used in Makar et al. [2] applies here. We also emphasize that our empirical results already operate in settings with multiple unknown shortcuts and demonstrate that both MBM and TIPMI remain robust when multiple shortcuts are present.
>
> **Citations**
>
> [1] Makar, Maggie, et al. "Causally motivated shortcut removal using auxiliary labels." International Conference on Artificial Intelligence and Statistics. PMLR, 2022.
>
> [2] Zheng, Jiayun, and Maggie Makar. "Causally motivated multi-shortcut identification and removal." Advances in Neural Information Processing Systems 35 (2022)

---

### Review · Reviewer_GaHH · 2025-10-28

**Summary Of Contributions:**

## Summary of contributions

The paper targets the problem of counterfactual invariance of a learned function, under the following assumptions: an anti-causal generative model $Y → M → X$with $X, Y$ potentially confounded, and $M$ mediator fully determined by $X$ and not affected by the confounder. Under these conditions, they show:

1. Given access to a dataset $(y_i, m_i, x_i)_i^n$ at training time, it is possible to learn a counterfactually invariant predictor $f: x \mapsto y$, where counterfactually invariant here means “unaffected by distribution shifts in $X$ due to confounders”.
2. The main contribution is algorithmic: the authors present a two-stage learning procedure, where they learn to predict $Y$ from $M$ (a teacher model) and $M$ from $X$ (a student model) at training time. At test time, this results in a function that can predict $X \mapsto M \mapsto Y$ and is counterfactually invariant. They present empirical evidence on three datasets

---

## Strengths
The presentation and the setup are clear. The empirical results, although they could be expanded, are convincing. Moreover, if the assumptions of the data-generating process are satisfied, the proposed method seems to successfully mitigate the issues of previous algorithms that targeted shortcut learning, namely: (i) the need for a priori knowledge on the possible shortcuts, clearly unrealistic, and (ii) the need of inefficient optimization constraints to enforce invariances.

---
## Weaknesses

There is one key assumption, that the relation between $X$ and $M$ is deterministic with $M$ unaffected by the confounder, that makes the learning problem somewhat trivial. I have nothing against the simplicity of the derived conclusion, but further discussion is needed to understand (i) the relation of the proposed setup with the (causality) literature (ii) its relevance.

**The relation with the *frontdoor* setup**

The paper’s graph and learning recipe are best read as an **anti-causal front-door** setup: $Y \to M \to X$ with shortcuts (i.e. latent confounder) $Y \leftarrow V \to X$. We already know from the frontdoor criterion that an unconfounded mediator allows estimation of the causal effect. A reinterpretation of this, is that a mediator allows to “disentangling” spurious relations (in this setting, latent shortcuts) from causal information (in this case, information flowing in the anticuasla direction from $X$ to $Y$). Seen through this lens, the conclusions of this paper aren’t surprising. I think it is needed to discuss this point in the paper, to clarify that the novelty in the paper is in the algorithm and the empirical result, while there is no conceptual/technical contribution in the choice of using a mediator to distill causal information from dependencies due to latent confounders.

**Stronger assumptions—why are they sensible for shortcut learning?**
The core theoretical guarantees hinge on assumptions that make the learning problem comparatively easy: $M$ must (i) carry all task-relevant information from $Y$ to $X$, (ii) be immune to shortcut variation $V$, and (iii) be (near-)deterministically retrievable from $X$. These are stronger than standard front-door conditions. The paper should motivate and defend them in the context of shortcut learning. From the paper, I don’t understand why this is a sensible modeling choice of relevant problems. Why do authors believe that we have access to mediators that shield the effect of latent confounders?

**More experiments**. Another concern that I have is that experiments on three datasets are not enough to derive “statistically” significant conclusions. I praise the authors for relying on real datasets for their main experiments; however, it would be good to have additional (real or synthetic) results to back their claim.

**Audience:**

Yes

**Audience Explanation:**

Yes. The authors propose a conceptually and empirically valid solution to the problem of shortcut learning, a problem of well-known interest. Also, their result is well placed in continuity with previous literature (Veicht et al. (2021), Makar et al. (2022))

**Broader Impact Concerns:**

I don't have any broader impact concern.

**Claims And Evidence:**

Yes

**Claims Explanation:**

The shortcut learning problem is well known and studied. The proposed solution (use of a mediator) is convincing: if we interpret the graph as casual, it reminds of the front door criterion setup, where you can "disentangle" causal information from the dependencies due to spurious confounders. The same idea is applied in this setup, and I believe it to be reasonable. The empirical evidence provided by the authors supports their claim. The only caveat, is that more experiments are needed: conclusions are drawn on three datasets, which is not sufficient to generalize from them.

**Requested Changes:**

As specified in the summary of the contribution, I believe that

1. [Critical] The paper needs to relate the conceptual idea of using a mediator to bypass latent confounders with the front door criterion, as discussed in the weaknesses section of the summary of the paper.
2. [Critical] Also, experiments on three datasets only can not be regarded as sufficient evidence for the effectiveness of their method. Experiments on more datasets are needed, as discussed in the weaknesses section of the summary of the paper
3. [Minor] I get why the proposed solution leads to better performance. What I don't understand is why the proposed solution would be realistic. The authors mention Knee osteoarthritis datastes as a motivation, but I am not sure that is enough. For details on this please refer to the discussion in the weaknesses section of the summary of the paper

---

> ### Author Response · Authors · 2025-11-07
> **Front-door criterion, experiments, and assumptions**
>
> We thank the reviewer for their thoughtful and constructive feedback!
>
> **1 - Relating the work to the front-door criterion.**
>
> We now discuss the relation to the front-door criterion in Section 3.1 of the revised manuscript. Like the front-door criterion, our setup relies on a mediator to separate spurious associations due to latent confounders. However, our setup differs in several important ways.
>
> First, the front-door criterion addresses the problem of identifying causal effects. In contrast, we assume an anti-causal prediction setting where our goal is not to recover causal effects, but to obtain a counterfactually invariant predictive model.
>
> Secondly, there is a backdoor path from $X$ to $M$ in our setup, e.g., $X \rightarrow V \rightarrow Y \rightarrow M$, which means that learning $X \rightarrow M$ may not remove all spurious correlations. Nevertheless, we show that under our stated assumptions -- most importantly, that $M$ is fully recoverable from $X$ -- that MBM and TIPMI can still yield a predictor that is invariant to shortcuts in the anti-causal regime.
>
> **2 - Additional experiments are needed.**
>
> We note that evaluating shortcut-robustness methods across three or fewer datasets is standard practice in this area (e.g., Makar et al. [1], Sagawa et al. [2], and Veitch et al. [3]). Consistent with prior work, we evaluated our methods across three diverse datasets that span natural images (Waterbirds), medical images (KOA), and text (Food Review). The observed improvements across all three datasets provide consistent empirical support  for our claims, and we believe the breadth of these datasets is appropriate for this setting.
>
> **3 - Why are the assumptions sensible for shortcut learning?**
>
> We appreciate the reviewer's question regarding the realism of our assumptions. We agree this is an important point, and we have clarified the motivation in the revised manuscript.
>
> First, mediators that are not causally affected by shortcuts are often available in practice. For instance, in the KOA setting, we use joint-space width measurements of the knee. Similar annotated anatomical structures are often available for other medical-imaging tasks. More broadly, in vision applications, segmentations or structured representations of the relevant object or region frequently serve as mediators and are not influenced by spurious shortcuts, e.g., background artifacts.
>
> Second, we acknowledge that mediators may be imperfect in practice. To address this, we added new experiments (see Appendix F) evaluating MBM and TIPMI when the shortcut influences the mediator in the KOA dataset. Our results show that both methods can remain more robust than standard L2 regularization as long as the influence of the shortcut is moderate.
>
> Finally, we have revised Section 3.1 and the introduction to provide a more detailed discussion of the assumption that mediators are fully recoverable from the input feature $\boldsymbol{X}$. This assumption follows prior work in robustness and causally motivated methods (e.g., Makar et al. [1]), where mediators are modeled as deterministic functions of the inputs to enable clear theoretical analysis for the proof of Proposition 1. We also note that this assumption is typically satisfied in settings where the input features are rich, unstructured data, such as image or text data.
>
> **Citations**
>
> [1] Makar, Maggie, et al. "Causally motivated shortcut removal using auxiliary labels." International Conference on Artificial Intelligence and Statistics. PMLR, 2022.
>
> [3] Sagawa, Shiori, et al. "Distributionally Robust Neural Networks." International Conference on Learning Representations.
>
> [4] Veitch, Victor, et al. "Counterfactual invariance to spurious correlations in text classification." Advances in neural information processing systems 34 (2021): 16196-16208.

---

### Decision · Action_Editor_H2bn · 2025-12-22

**Recommendation:** Accept with minor revision

**Audience:**

Yes

**Audience Explanation:**

This paper speaks to two community, the causality one and researchers interested in out-of-distribtion in ML. As such I would say it is of broad interest for several sub-communities in TMLR's audience.

**Claims And Evidence:**

Yes

**Claims Explanation:**

This paper tackles the problem of out-of-distribution generalization using causal models, in particular, mediation information. The reviewers praised the presentation and setup, as well as the experimental evidence. Overall, the reviewers had consensus that the paper should be accepted at TMLR as this is an improtant problem as the approach of the paper is sound both on the theory and experimentation side.

The only outstanding question is:

```
I think it is a good idea to work with two confounders for the main part of the paper. However, this is not the case for the theoretical results. Since you "stress that [your] results hold for V of arbitrary dimension" in Section 3.1, I would like to see that proven in the paper. If there are already extensions to the analysis you base your results on for more confounders, then that should be all the more reason to state your results for more than two confounders. The same holds for your remark on the experimental setting. If your current results that are stated for two confounders do not apply in your experimental setting, this is all the more reason to prove your results in the more general setting.
```
**Update request:** I would kindly ask the authors to either address this, or call this out as a limitation for future work.